# NEURAL BRIDGE PROCESSES

## ABSTRACT

Learning stochastic functions from partially observed context-target pairs is a fundamental problem in probabilistic modeling. Traditional models like Gaussian Processes (GPs) face scalability issues with large datasets and assume Gaussianity, limiting their applicability. While Neural Processes (NPs) offer more flexibility, they struggle with capturing complex, multi-modal target distributions. Neural Diffusion Processes (NDPs) enhance expressivity through a learned diffusion process but rely solely on conditional signals in the denoising network, resulting in weak input coupling from an unconditional forward process and semantic mismatch at the diffusion endpoint. In this work, we propose Neural Bridge Processes (NBPs), a novel method for modeling stochastic functions where inputs $x$ act as dynamic anchors for the entire diffusion trajectory. By reformulating the forward kernel to explicitly depend on $x$, NBP enforces a constrained path that strictly terminates at the supervised target. This approach not only provides stronger gradient signals but also guarantees endpoint coherence. We validate NBPs on synthetic data, EEG signal regression and image regression tasks, achieving substantial improvements over baselines. These results underscore the effectiveness of DDPM-style bridge sampling in enhancing both performance and theoretical consistency for structured prediction tasks.

## 1 INTRODUCTION

Learning stochastic functions from partially observed context-target pairs is a fundamental problem in probabilistic modeling Rasmussen (2003); Garnelo et al. (2018b;a); Dutordoir et al. (2023); Franzese et al. (2023); Bonito et al. (2023); Mathieu et al. (2023); Chang et al. (2024); Dou et al. (2025); Hamad & Rosenbaum, playing a pivotal role in meta-learning Garnelo et al. (2018b;a), few-shot regression Kim et al. (2019), Bayesian optimization Dutordoir et al. (2023); Krishnamoorthy et al. (2023), and uncertainty-aware prediction tasks Chang et al. (2024); Dou et al. (2025); Hamad & Rosenbaum; Requeima et al. (2024). Such problems require models that not only generalize well across different tasks but also provide calibrated uncertainty estimates, particularly under scarce or incomplete data conditions. Gaussian Processes (GPs) Rasmussen (2003) have traditionally dominated this area due to their analytical tractability and clear uncertainty quantification. However, GPs inherently assume Gaussianity and exhibit cubic computational complexity with respect to data size, severely limiting their applicability in scenarios involving large datasets or inherently non-Gaussian functional distributions Snelson & Ghahramani (2005); Titsias (2009); Xu & Zeng (2024); Xu et al. (2024).

Neural Processes (NPs) Garnelo et al. (2018b;a); Kim et al. (2019); Louizos et al. (2019) have emerged as a compelling alternative, merging the flexibility of neural network models with the principled uncertainty quantification of stochastic processes. By parameterizing stochastic functions through neural architectures, NPs facilitate efficient inference and scalable learning, successfully tackling meta-learning and few-shot prediction tasks. However, standard NPs often struggle with limited expressivity and fail to capture complex multi-modal target distributions, motivating exploration into more powerful generative mechanisms.

NDPs Dutordoir et al. (2023) address this by modeling the input-output mapping as a learned diffusion process Ho et al. (2020), offering enhanced expressivity and sample diversity. Despite their promise in stochastic function modeling, NDPs rely on an unconditional forward process, which fundamentally limits the effectiveness of input supervision. Specifically, traditional NDPs treat inputs merely as conditional signals within the denoising network, passively injecting inputs during denoising without

leveraging the temporal structure of diffusion. This results in weak input coupling and a semantic mismatch at the diffusion endpoint.

In this work, we introduce Neural Bridge Processes (NBP), a novel diffusion-based generative framework that explicitly integrates input supervision throughout the entire diffusion trajectory. Unlike traditional NDPs, which inject conditioning inputs passively during denoising, NBPs reformulate the forward diffusion kernel to dynamically anchor the process with inputs $x$, ensuring that the generated outputs remain coherently guided toward the desired targets. This is achieved through a principled bridge coefficient $\gamma_t$, which progressively strengthens the influence of $x$ as diffusion proceeds, enabling both strong gradient signals during training and guaranteed endpoint coherence. Additionally, NBPs incorporate a bridge correction term in the reverse process to maintain theoretical consistency between forward and reverse dynamics. Our approach provides a more structured and controllable generative path, leading to improved conditional generation accuracy and more faithful reconstructions, particularly in settings requiring strict adherence to input-output relationships. This idea shares conceptual similarities with recent advances in diffusion bridge modeling Zhou et al. (2023); Yue et al. (2023); Zheng et al. (2024); Li et al. (2023); Peluchetti (2023); He et al. (2024); Shi et al. (2023); Naderiparizi et al. (2025). However, our approach avoids the more complex and computationally intensive SDE-style diffusion bridges Song et al. (2020); Zhou et al. (2023), and instead extends the bridge concept to the DDPM Ho et al. (2020) framework through SNR-aware functional modeling, making it significantly easier to deploy and integrate into existing architectures.

We validate NBPs on synthetic data and real time series data consisting of electroencephalogram (EGG) measurements Zhang et al. (1995) and image regression tasks, achieving substantial improvements over baseline NDPs. These results underscore the effectiveness of DDPM-style bridge sampling in enhancing both performance and theoretical consistency for structured prediction tasks.

In summary, the core contributions of this paper are:

- We introduce **Neural Bridge Processes (NBPs)**, a new class of models for stochastic functions that introduces input-anchored diffusion trajectories via a principled bridge coefficient. This design ensures strong input supervision throughout the entire diffusion process, overcoming the weak coupling limitations of traditional NDPs and guaranteeing endpoint coherence.

- We extend the bridge concept to the DDPM framework using an SNR- and path-aware formulation, thereby avoiding the deployment complexity associated with SDE-based diffusion bridges. This makes NBPs both theoretically consistent and practically efficient, enabling seamless integration into existing architectures.

- We demonstrate the effectiveness of NBPs on synthetic data, EEG signal regression and image-based function regression benchmarks, achieving significant improvements in predictive accuracy and uncertainty calibration compared to state-of-the-art NDP baselines.

## 2 BACKGROUND: NEURAL PROCESSES

Neural Processes (NPs) Garnelo et al. (2018b;a); Kim et al. (2019); Louizos et al. (2019) combine the expressiveness of neural networks with the probabilistic reasoning of Gaussian Processes (GPs) Rasmussen (2003). While GPs offer principled uncertainty quantification, they suffer from poor scalability Snelson & Ghahramani (2005); Titsias (2009) and limited kernel flexibility Wilson et al. (2016); Liu et al. (2021). In contrast, Neural Networks (NNs) Schmidhuber (2015); Nielsen (2015) are highly flexible and scalable but lack inherent mechanisms for uncertainty modeling Blundell et al. (2015); Pearce et al. (2020); Gawlikowski et al. (2023). NPs address these limitations by modeling distributions over functions using a neural network-based framework. They approximate a stochastic process $F : X \rightarrow Y$ through finite-dimensional marginals, parameterized by a latent variable $z$ to capture global uncertainty. Given context observations $(x_{\mathbb{C}}, y_{\mathbb{C}})$ and target inputs $x_{\mathbb{T}}$, NPs generate predictive distributions over $y_{\mathbb{T}}$ via a conditional latent model.

$$p(y_{\mathbb{T}}, z | x_{\mathbb{T}}, x_{\mathbb{C}}, y_{\mathbb{C}}) = p(z | x_{\mathbb{C}}, y_{\mathbb{C}}) \prod_{i=1}^{|\mathbb{T}|} p(y_{\mathbb{T},i} | x_{\mathbb{T},i}, z) \tag{1}$$

Here, $z$ encodes the uncertainty about the global structure of the underlying function. Training NPs uses amortized variational inference, optimizing an evidence lower bound (ELBO) on the conditional log-likelihood:

$$\log p(y_\mathbb{T}|x_\mathbb{C}, y_\mathbb{C}, x_\mathbb{T}) \geq \mathbb{E}_{q(z|x_\mathbb{C}, y_\mathbb{C})} \left[ \sum_{i \in \mathbb{T}} \log p(y_{\mathbb{T},i}|z, x_{\mathbb{T},i}) + \log \frac{p(z|x_\mathbb{C}, y_\mathbb{C})}{q(z|x_\mathbb{C}, y_\mathbb{C})} \right] \tag{2}$$

where $q(z|x_\mathbb{C}, y_\mathbb{C})$ is the variational posterior distribution parameterized by a neural network, and $p(z|x_\mathbb{C}, y_\mathbb{C})$ is the conditional prior. Additional information can be seen in Appendix.

## 3 METHOD: NEURAL BRIDGE PROCESSES (NBP)

### 3.1 PROBLEM SETUP

We consider the standard meta-learning setting where a model observes a set of context points $x_\mathbb{C} = \{(x_i, y_i)\}_{i=1}^{N_c}$ and aims to predict the corresponding outputs $y_\mathbb{T} = \{y_j\}_{j=1}^{N_t}$ for a set of target inputs $x_\mathbb{T} = \{x_j\}_{j=1}^{N_t}$. Here, each task is assumed to be sampled from a distribution over functions, and the goal is to model the conditional distribution $p(y_\mathbb{T}|x_\mathbb{C}, x_\mathbb{T})$.

### 3.2 REVIEW: NEURAL DIFFUSION PROCESSES (NDPs)

While Neural Processes (NPs) Garnelo et al. (2018b) effectively combine neural networks with stochastic processes for few-shot learning, their reliance on simple latent variable models limits their ability to capture complex, multimodal distributions. Neural Diffusion Processes (NDPs) Dutordoir et al. (2023) address this by introducing stochastic trajectories, modeling the mapping from inputs to outputs as a learned diffusion process Ho et al. (2020). This method significantly improves expressivity and sample diversity by leveraging the generative power of diffusion models Ho et al. (2020); Song et al. (2020); Dhariwal & Nichol (2021); Croitoru et al. (2023); Rombach et al. (2022); Podell et al. (2023); Peebles & Xie (2023), allowing for better modeling of complex distributions and more flexible conditional sampling.

Formally, given a function $f : \mathbb{R}^D \rightarrow \mathbb{R}$, an NDP learns a generative distribution over observed data pairs $(x, y)$, where inputs $x \in \mathbb{R}^{N \times D}$ and outputs $y = f(x) \in \mathbb{R}^N$. Unlike standard NPs, NDPs Dutordoir et al. (2023) do not explicitly require a partitioning into context and target sets during training; all points are jointly modeled. In supervised learning setting, the NDP modeling framework consists of two stochastic processes:

**Forward Diffusion Process.** Starting from observed clean data $y_0$, the forward diffusion process gradually injects Gaussian noise into the outputs over $T$ timesteps according to a predefined variance schedule $\{\beta_t\}$:

$$q(y_{1:T} \mid y_0) = \prod_{t=1}^{T} q(y_t \mid y_{t-1}), \quad q(y_t \mid y_{t-1}) = \mathcal{N}(y_t; \sqrt{1 - \beta_t}\, y_{t-1}, \beta_t I). \tag{3}$$

After $T$ diffusion steps, the distribution of the outputs converges towards standard Gaussian noise, i.e., $y_T \sim \mathcal{N}(0, I)$.

**Reverse Process.** Neural Diffusion Processes (NDPs) learn a conditional reverse process that denoises observations from Gaussian noise $y_T$ to outputs $y_0$, guided by an input $x$:

$$p_\theta(y_{0:T} \mid x) = p(y_T) \prod_{t=1}^{T} p_\theta(y_{t-1} \mid y_t, x), \tag{4}$$

with Gaussian transitions parameterized by a noise prediction model $\epsilon_\theta$:

$$p_\theta(y_{t-1} \mid y_t, x) = \mathcal{N}\left(y_{t-1}; \mu_\theta(y_t, t, x), \tilde{\beta}_t I\right). \tag{5}$$

where $\tilde{\beta}_t = \frac{1-\bar{\alpha}_{t-1}}{1-\bar{\alpha}_t}\beta_t$, $\mu_\theta(y_t, t, x)$ is reparameterized as

$$\mu_\theta(y_t, t, x) = \frac{1}{\sqrt{1-\beta_t}}\left(y_t - \frac{\beta_t}{\sqrt{1-\bar{\alpha}_t}}\epsilon_\theta(y_t, t, x)\right), \quad \bar{\alpha}_t = \prod_{s=1}^{t}(1-\beta_s) \qquad (6)$$

**Training Objective.** NDPs employ a denoising score matching objective Hyvärinen & Dayan (2005); Song et al. (2021); Huang et al. (2021), training the noise model $\epsilon_\theta$ by minimizing the discrepancy between predicted noise and actual noise $\epsilon \sim \mathcal{N}(0, I)$:

$$\mathcal{L}_\theta = \mathbb{E}_{t,x,y_0,\epsilon}\left[\|\epsilon - \epsilon_\theta(y_t, t, x)\|_2^2\right], \quad \text{with} \quad y_t = \sqrt{\bar{\alpha}_t}y_0 + \sqrt{1-\bar{\alpha}_t}\epsilon. \qquad (7)$$

## 3.3 MOTIVATION

Traditional Neural Diffusion Processes (NDPs) treat inputs merely as conditional signals fed into the denoising network (e.g., via cross-attention or concatenation). This implicit conditioning suffers from two critical drawbacks:

- Weak coupling: The diffusion path is only loosely guided by inputs, as the forward process remains an unconditional Gaussian transition $q(y_t|y_{t-1})$.
- Endpoint mismatch: The diffusion endpoint $y_T$ is arbitrary noise, bearing no semantic relationship to the input supervision $x$.

While NDPs have shown promise in generative modeling, their unconditional forward process fundamentally limits the efficacy of input supervision. Existing methods inject inputs passively during denoising, failing to exploit the temporal structure of diffusion. In this work, we propose Neural Bridge Processes (NBP), where inputs $x$ act as dynamic anchors for the entire diffusion trajectory. By reformulating the forward kernel to explicitly depend on $x$, NBP enforces a constrained path that strictly terminates at the supervised target. This approach not only provides stronger gradient signals but also guarantees endpoint coherence—a property unattainable by traditional NDPs.

## 3.4 BRIDGE CONSTRUCTION

We first consider the case where the input and output share the same dimensionality. Given a function $f : \mathbb{R}^D \to \mathbb{R}^D$, our Neural Bridge Processes (NBPs) model a generative distribution over observed data pairs $(x, y)$, with inputs $x \in \mathbb{R}^{N \times D}$ and outputs $y = f(x) \in \mathbb{R}^{N \times D}$. NBPs construct a diffusion bridge between arbitrary initial outputs $y_0 = y$ and conditionally anchored endpoints characterized by the relationship between $\mathbb{E}[y_T]$ and $x$ via modified transition kernels.

To achieve this, we introduce a time-dependent coefficient $\gamma_t$ that explicitly controls the influence of the input $x$ on the forward diffusion process at each timestep $t$, thereby enabling explicit path supervision in contrast to standard DDPMs.

The forward transition kernel is defined as:

$$q(y_t|y_{t-1}, x) = \mathcal{N}\left(y_t; \underbrace{\sqrt{1-\beta_t}y_{t-1} + \gamma_t x}_{\text{Bridge-anchored mean}}, \beta_t I\right). \qquad (8)$$

The bridge coefficient $\gamma_t$ follows a principled design:

$$\gamma_t = \frac{\text{SNR}_T}{\text{SNR}_t}, \quad \text{SNR}_t = \frac{\bar{\alpha}_t}{1-\bar{\alpha}_t}. \qquad (9)$$

Similar to DDPM, the forward process in this bridge-style diffusion model allows sampling $y_t$ at an arbitrary timestep $t$ in closed form:

$$y_t \mid y_0, x \sim \mathcal{N}\left(\sqrt{\bar{\alpha}_t}\, y_0 + \bar{\gamma}_t x, \ (1-\bar{\alpha}_t)I\right), \qquad (10)$$

where the cumulative bridge coefficient is defined as

$$\bar{\gamma}_t = \sum_{s=1}^{t}\gamma_s\sqrt{\frac{\bar{\alpha}_t}{\bar{\alpha}_s}}. \qquad (11)$$

The term $\gamma_t x$ in Equation (8) acts as a guiding force that progressively pulls the diffusion trajectory toward the target endpoint. This formulation ensures the following behavior:

- **Early Diffusion Phase** ($t \ll T$): $\mathrm{SNR}_t \to +\infty \Rightarrow \gamma_t \to 0^+$. Process approximates standard diffusion:

$$q(y_t|y_{t-1}, x) \approx \mathcal{N}(\sqrt{1 - \beta_t}y_{t-1}, \beta_t I) \tag{12}$$

- **Bridge Convergence Phase** ($t \to T$): $\mathrm{SNR}_t \to \mathrm{SNR}_T \Rightarrow \gamma_t \to 1$. The trajectory is increasingly guided toward the desired target to enforces endpoint attraction as shown in Equation (10):

$$\mathbb{E}[y_T|y_0] = \sqrt{\bar{\alpha}_T}y_0 + \bar{\gamma}_T x \approx \bar{\gamma}_T x \tag{13}$$

where $\bar{\alpha}_T = \prod_{s=1}^{T}(1 - \beta_s) \approx 0$ , and $\bar{\gamma}_T = \left(\sum_{s=1}^{T} \gamma_s \sqrt{\frac{\bar{\alpha}_T}{\bar{\alpha}_s}}\right)$ is a constant.

This introduces:

1). Stronger gradient signals: The bridge term $\gamma_t x$ directly propagates input supervision to every timestep of the forward process by Equation (8) and (10), thereby introducing $x$ into the denoising network in a more theoretically consistent manner.

2). Coherent trajectory optimization: The entire path $y_{0:T}$ is trained to satisfy both data fidelity (to $y_0$) and endpoint matching (to $\mathbb{E}[y_T] = \bar{\gamma}_T x$).

**Forward Process** Given the starting point $y_0$ and target endpoint $y_T$, the forward process progressively adds noise and enforces the bridge constraint through the transition kernel:

$$q(y_{1:T} \mid y_0, x) = \prod_{t=1}^{T} q(y_t \mid y_{t-1}, x) \tag{14}$$

The single-step transition kernel is defined in Equation (8). After $T$ diffusion steps, the distribution of the outputs converges to a Gaussian with mean $x$, i.e., $y_T \sim \mathcal{N}(\bar{\gamma}_T x, I)$.

**Reverse Process with Bridge Correction** The reverse process employs a transition kernel that combines standard denoising with explicit bridge constraints to maintain consistency with the forward process:

$$p_\theta(y_{t-1}|y_t, x) = \mathcal{N}\left(y_{t-1}; \mu_\theta(y_t, x, t), \tilde{\beta}_t I\right) \tag{15}$$

The mean function is reparameterized into two key components that balance denoising and bridge correction:

$$\mu_\theta = \underbrace{\frac{1}{\sqrt{\alpha_t}}\left(y_t - \frac{\beta_t}{\sqrt{1 - \bar{\alpha}_t}}\epsilon_\theta(y_t, x, t)\right)}_{\text{Denoising term}} + \underbrace{C_t(x)}_{\text{Bridge correction}} \tag{16}$$

where $\alpha_t = 1 - \beta_t$, and the bridge correction term $C_t(x)$ is derived as:

$$C_t(x) = -\frac{\gamma_t}{\sqrt{1 - \beta_t}}x. \tag{17}$$

Proofs can be seen in Appendix. The role of the bridge correction term $C_t(x)$ is to ensure that the mean of the reverse process remains consistent with the bridge constraint imposed in the forward process. Specifically, in the forward process, the term $\gamma_t x$ injects information from the input $x$ into the diffusion trajectory. The reverse process compensates for this influence through $C_t(x)$.

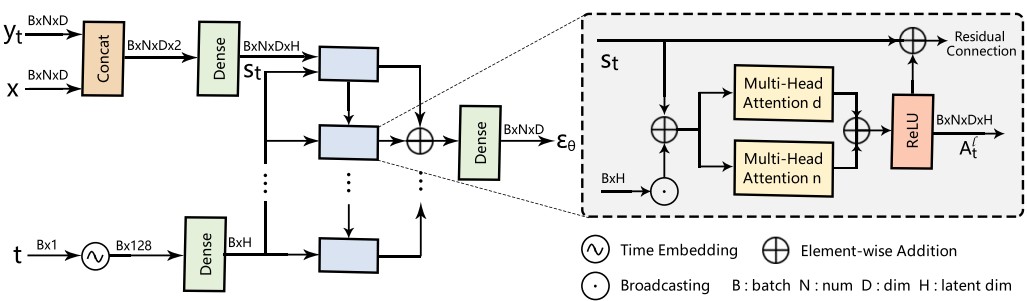

Figure 1: The noise model architecture employed at each step of the Neural Bridge Processes. The greyed-out box highlights the bi-dimensional attention block.

**Training Objective**   The training objective for NBPs minimizes the forward and reverse KL divergences, which is equivalent to minimizing the prediction error of the denoising network $\epsilon_\theta(y_t, x, t)$ and the forward noise characterized by Equation (10).

$$\mathcal{L}_\theta = \mathbb{E}_{t,x,y_0,\epsilon} \left[ \|\epsilon_\theta(y_t, x, t) - \epsilon\|_2^2 \right] \quad \text{with} \quad y_t = \sqrt{\bar{\alpha}_t} y_0 + \bar{\gamma}_t x + \sqrt{1 - \bar{\alpha}_t} \epsilon. \tag{18}$$

Proofs can be seen in Appendix. Here, $\epsilon \sim \mathcal{N}(0, I)$ represents the ground-truth noise added during the forward process. The reverse transition kernel's dependence on both $y_t$ and $x$ ensures theoretical consistency, as the bridge variable $x$ is explicitly embedded in both the forward and reverse processes. This formulation guarantees that the learned reverse process remains properly coupled with the forward dynamics throughout the diffusion trajectory.

**Conditional Sampling Procedure.**   At test time, NBPs generate samples from the conditional distribution $p(y_0 \mid x, \mathcal{D})$, where $\mathcal{D} = (x_\mathbb{C} \in \mathbb{R}^{M \times D}, y_{\mathbb{C},0} \in \mathbb{R}^{M \times D})$ denotes the observed context data.

The conditional sampling proceeds as follows. First, initialize the diffusion state with the known endpoint:

$$y_T = \gamma_T x + n, \quad n \sim \mathcal{N}(0, I). \tag{19}$$

For each diffusion timestep $t = T, \ldots, 1$, perform the following steps:

- Sample the noisy context outputs using the forward diffusion bridge process:

$$y_{\mathbb{C},t} \sim \mathcal{N}\left( \sqrt{\bar{\alpha}_t} y_{\mathbb{C},0} + \bar{\gamma}_t x_\mathbb{C}, (1 - \bar{\alpha}_t)I \right), \tag{20}$$

- Combine noisy target and context states at timestep $t$:

$$y_t = \{y_{\mathbb{T},t}, y_{\mathbb{C},t}\}, \quad x = \{x_\mathbb{T}, x_\mathbb{C}\}. $$

- Perform the reverse diffusion step with the learned backward kernel, incorporating the bridge correction:

$$y_{t-1} \sim \mathcal{N}\left( \mu_\theta(y_t, x, t), \tilde{\beta}_t I \right), \tag{21}$$

  where

$$\mu_\theta(t) = \frac{1}{\sqrt{\alpha_t}} \left( y_t - \frac{\beta_t}{\sqrt{1 - \bar{\alpha}_t}} \epsilon_\theta(y_t, x, t) \right) + C_t(x). \tag{22}$$

Following the Repaint Lugmayr et al. (2022) strategy, at each diffusion timestep $t$, we repeat the forward perturbation of context points and the corresponding reverse denoising step multiple times before proceeding to the next timestep. Simulating this repeated scheme from $t = T$ down to $t = 1$ ensures that the context information is consistently reinforced throughout the diffusion trajectory, leading to more coherent and accurate conditional generation.

### 3.5 INPUT-OUTPUT DIMENSIONAL ALIGNMENT

Real-world datasets often have mismatched input and output dimensions. To enable joint modeling in the same space, we apply a fixed projection $\mathcal{P}$ to map $x \in \mathbb{R}^{N \times D_x}$ to $x_a = \mathcal{P}(x) \in \mathbb{R}^{N \times D_y}$, aligning it with $y \in \mathbb{R}^{N \times D_y}$. For example, in image regression, $\mathcal{P}$ can add spatial or contextual information to match RGB output dimensions.

### 3.6 NOISE MODEL ARCHITECTURE

To ensure that our model remains consistent with the structural properties of stochastic processes and to guarantee fair experimental comparisons, we adopt the same noise model architecture as NDPs, namely the Bi-Dimensional Attention Block Dutordoir et al. (2023), as shown in Figure 1. Due to space limitations, we provide the detailed design in the Appendix.

## 4 EXPERIMENTS

### 4.1 BASELINE IMPLEMENTATION AND EVALUATION METRICS

For a comprehensive comparison, we implement Neural Processes (NPs) Garnelo et al. (2018b), Attentive Neural Processes (ANPs) Kim et al. (2019), and Convolutional Neural Processes (ConvNPs) Gordon et al. (2019) using the official NP-Family repository Dubois et al. (2020), with all hyperparameters set to the recommended defaults. We further include Gaussian Neural Processes (GNPs) Bruinsma et al. (2021) in our synthetic experiments. For Neural Diffusion Processes (NDPs) Dutordoir et al. (2023), Geometric Neural Diffusion Processes (GEOMNDPs) Mathieu et al. (2023), and Score-Based Neural Processes (SNPs) Dou et al. (2025), we directly adopt their official implementations. To ensure fairness, our Neural Bridge Process (NBP) employs the same Bi-Dimensional Attention Block architecture and hyperparameter configurations as the baseline NDP. Detailed implementation settings and additional related work are provided in the Appendix. All models are retrained on the experimental datasets for consistent metric evaluation and visualization. Experiments are conducted on a single NVIDIA RTX 4090 GPU.

### 4.2 REGRESSION ON SYNTHETIC DATA

We evaluate our method on synthetic 1D–3D regression tasks, using functions sampled from Gaussian Processes (GPs) with either a Squared Exponential or Matérn-5/2 kernel. For each dimension $D$, the kernel lengthscale is set to $\ell = \sqrt{D}/4$, and Gaussian noise $\mathcal{N}(0, 0.05^2)$ is added to the outputs. During training, we generate $2^{10}$ examples per epoch, and train for 400 epochs using batch size 32. Each model is trained with its own architecture and optimization settings (details below). At test time, the context set contains a random number of points between 1 and $10 \times D$, while the target set always includes 50 points.

The log-likelihood is estimated by fitting a multivariate Gaussian to 128 samples drawn from the conditional distribution of the model. For our proposed method, we use a 4-layer transformer-style architecture with 8 attention heads and 64-dimensional hidden layers. Diffusion noise is scheduled over 500 timesteps with a cosine schedule ($\beta \in [3e-4, 0.5]$). The optimizer uses a peak learning rate of $10^{-3}$ with warm-up (20 epochs) and cosine decay (200 epochs). All experiments use the same evaluation batch size and sampling procedure for consistency. For input dimensions $D = 2, 3$, to address the alignment issue discussed in Section 3.5, we directly define the projection operator $\mathcal{P}$ in Equation (20) as the mean of the input components. Meanwhile, the input $x$ to the denoising network $\epsilon_\theta$ remains unchanged.

Table 2 shows that NBP consistently outperforms prior Neural Process variants across all input dimensions. Notably, NBP maintains stable and accurate predictions in higher dimensions ($D = 2, 3$), where performance of other models tends to degrade sharply. Figure 4 shows representative samples generated by our model under the Squared Exponential (SE) kernel and the Matérn-5/2 kernel settings.

### 4.3 REAL WORLD EXPERIMENTS

#### 4.3.1 EEG SIGNAL REGRESSION TASKS

We evaluate the proposed Neural Bridge Processes (NBP) on a real-world electroencephalogram (EEG) dataset regression task Zhang et al. (1995). The dataset comprises 7632 multivariate time series, each consisting of 256 evenly sampled time steps recorded across seven electrode channels. These EEG signals show strong temporal dynamics and cross-channel correlations, making them ideal for evaluating multi-output meta-learning models like NBPs.

To assess NBPs on correlated multi-output prediction and missing data, we randomly mask windows in 3 of 7 channels and predict the missing values. Inputs are the concatenated temporal and channel indices $\mathbf{x}_e = (i_t, i_c)$, and outputs are the voltage measurements $\mathbf{y}_e$. This evaluation is carried out under three distinct experimental settings:

- **Interpolation**: Predicting missing values within the existing temporal span.
- **Reconstruction**: Predicting values from partially obscured temporal segments.
- **Forecasting**: Predicting future values beyond the observed temporal data.

Performance metrics employed include Mean Squared Error (MSE) and Negative Log-Likelihood (NLL). As demonstrated in Table 1, the NBP consistently surpasses baseline methods across all three scenarios, underscoring NBPs' proficiency in capturing intricate temporal structures and cross-channel dependencies inherent in EEG data.

Table 1: Predictive NLL ($\downarrow$) and MSE ($\downarrow$) on EEG

| Method | Inter. | | Recon. | | Forec. | |
|---|---|---|---|---|---|---|
| | NLL | MSE($\times 10^{-2}$) | NLL | MSE($\times 10^{-2}$) | NLL | MSE($\times 10^{-2}$) |
| NP | 1.66 | 0.52 | 1.78 | 0.44 | 1.61 | 0.39 |
| ANP | 0.47 | 0.25 | 0.70 | 0.48 | 0.90 | 0.60 |
| ConvNP | 0.44 | 0.40 | -2.43 | 0.40 | -2.34 | 0.55 |
| NDP | -2.46 | 0.18 | -2.59 | 0.23 | -2.69 | 0.38 |
| SNP | -3.19 | 0.16 | **-3.30** | 0.18 | -3.02 | 0.31 |
| GEOMNDP | -2.48 | 0.18 | -2.65 | 0.20 | -2.84 | 0.34 |
| **NBP (Ours)** | **-3.35** | **0.16** | -3.22 | **0.16** | **-3.51** | **0.29** |

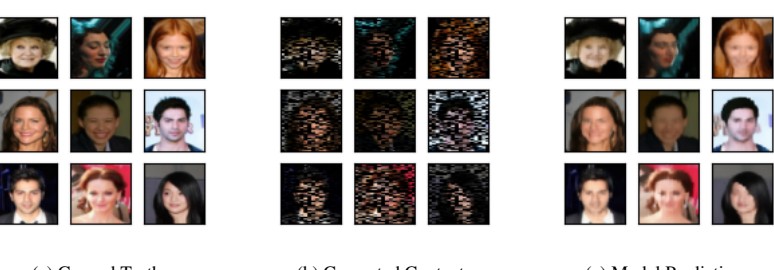

(a) Ground Truth          (b) Corrupted Context          (c) Model Prediction

Figure 2: Qualitative Results on CelebA $64 \times 64$ Image Regression Task: (a) Ground Truth, (b) Corrupted Context, (c) Model Prediction. The task requires inferring the true image content based on randomly corrupted context information and the spatial coordinates of target pixels.

#### 4.3.2 IMAGE REGRESSION TASK

In this experiment, we apply Neural Bridge Processes (NBPs) to the image regression task, where the objective is to predict pixel values based solely on their spatial coordinates normalized within the range $[-2, 2]$. We conduct experiments using the CelebA dataset at resolutions of $32 \times 32$ and $64 \times 64$. The experimental setup, including hyperparameters, denoising network architecture, learning rate, random seed, and other baseline configurations, strictly follows the standard configuration used by NDPs for a fair comparison.

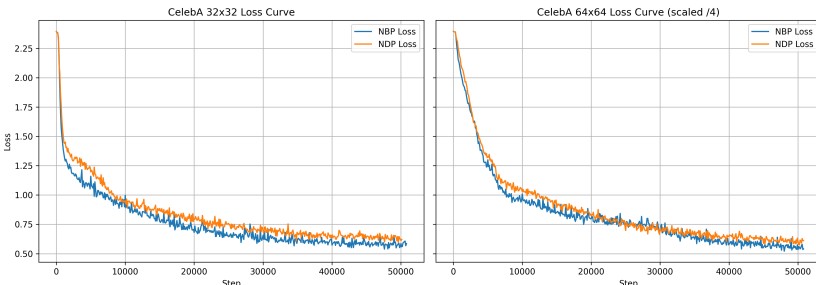

Figure 3: Comparison of NBP and NDP: NBPs consistently achieve lower loss values across the majority of training iterations

We evaluate the performance of NBPs under different levels of context sparsity and directly compare their predictive accuracy with that of NDPs. The Mean Squared Error (MSE) results are summarized in Figure 5, with detailed numerical values provided in Tables 4 and 5 in the Appendix. As shown in Figure 5, the proposed NBP (orange) consistently achieves lower MSE across all context ratios compared to the NDP baseline (blue). Moreover, NBPs exhibit smaller standard deviations, indicating more stable and reliable predictions. All MSE values are computed by averaging over nine conditional samples for each input, with pixel values normalized to the range $[0, 1]$. Figure 2 provides an illustrative depiction of the image regression task setup.

Under various observation conditions, NBPs consistently outperform NDPs by a substantial margin. For example, at a context ratio of $0.02$, NBPs achieve an MSE of $0.76$ compared to $0.88$ by NDPs on CelebA $32 \times 32$. This demonstrates NBPs' enhanced capability to capture spatial dependencies and deliver accurate predictions even from highly sparse context observations. This advantage extends to higher resolutions: at the same context ratio ($0.02$) on CelebA $64 \times 64$, NBPs achieve an MSE of $0.80$ compared to NDPs' $1.05$, underscoring the scalability and robustness of our method. Furthermore, Figure 3 visualizes the denoising score matching objective loss during training, illustrating that NBPs consistently achieve lower loss values across the majority of training iterations. This result supports the conclusion that the bridge-based training paradigm significantly enhances the effectiveness of denoising diffusion probabilistic model (DDPM) path supervision.

### 4.4 Computational Efficiency

We observe in our experiments that, under consistent training settings—including the denoising network architecture, learning rate, random seed, and other base hyperparameters—the proposed NBP model and the baseline NDP consume approximately the same amount of time per epoch and in total. This indicates that NBP does not introduce additional computational overhead. This efficiency stems from the fact that NBP does not incorporate any extra architectural complexity. Instead, it enhances the training signal through a coupling mechanism between the inputs and outputs in the neural diffusion process (as described in Section 3). This coupling is implemented entirely at the software level, without increasing the model's structural depth or parameter count.

### 5 Conclusion

In this work, we introduced Neural Bridge Processes (NBPs), a diffusion-based framework for stochastic function modeling that explicitly incorporates input supervision throughout the diffusion trajectory. By reformulating the forward kernel with a principled bridge coefficient, NBPs address the weak input coupling and endpoint mismatch of traditional Neural Diffusion Processes (NDPs), ensuring stronger conditional guidance and better theoretical consistency. Unlike computationally intensive SDE-based bridges, NBPs implement bridge corrections efficiently within the DDPM framework using SNR- and path-aware modeling. Experiments on synthetic data, real-world EEG time series, and image regression tasks demonstrate that NBPs significantly enhance predictive accuracy and uncertainty calibration compared to state-of-the-art NDP baselines. These results highlight the potential of NBPs for structured generative modeling, paving the way for future extensions to high-dimensional, multi-modal, and control-oriented applications.

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

# A    TABLES AND FIGURES IN THE MAIN TEXT

Table 2: Mean test log-likelihood ($\uparrow$) $\pm$ 1 standard error estimated over 128 test samples.

| | Squared Exponential | | | Matérn-$\frac{5}{2}$ | | |
|---|---|---|---|---|---|---|
| **Model** | $D = 1$ | $D = 2$ | $D = 3$ | $D = 1$ | $D = 2$ | $D = 3$ |
| ANP | $-4.79\pm0.05$ | $-23.80\pm0.05$ | $-23.80\pm0.04$ | $-0.70\pm0.04$ | $-17.22\pm0.02$ | $-21.24\pm0.02$ |
| ConvCNP | $-6.40\pm0.07$ | $-24.00\pm0.03$ | $-23.80\pm0.02$ | $-0.87\pm0.06$ | $-17.50\pm0.03$ | $-21.24\pm0.02$ |
| GNP | $4.00\pm0.02$ | $-19.60\pm0.02$ | $-23.80\pm0.02$ | $\mathbf{0.14}\pm0.02$ | $-15.70\pm0.02$ | $-21.20\pm0.02$ |
| NDP | $4.21\pm0.04$ | $-13.39\pm0.05$ | $-20.48\pm0.05$ | $-0.13\pm0.02$ | $-14.74\pm0.03$ | $-20.66\pm0.05$ |
| SNP | $4.27\pm0.02$ | $-13.19\pm0.03$ | $-20.24\pm0.04$ | $0.01\pm0.02$ | $-14.67\pm0.03$ | $-20.59\pm0.05$ |
| GEOMNDP | $4.22\pm0.05$ | $-13.36\pm0.05$ | $-20.45\pm0.05$ | $-0.13\pm0.03$ | $-14.73\pm0.03$ | $-20.63\pm0.06$ |
| **NBP (ours)** | $\mathbf{4.33}\pm0.03$ | $\mathbf{-13.15}\pm0.05$ | $\mathbf{-20.11}\pm0.04$ | $-0.05\pm0.02$ | $\mathbf{-14.62}\pm0.03$ | $\mathbf{-20.51}\pm0.03$ |

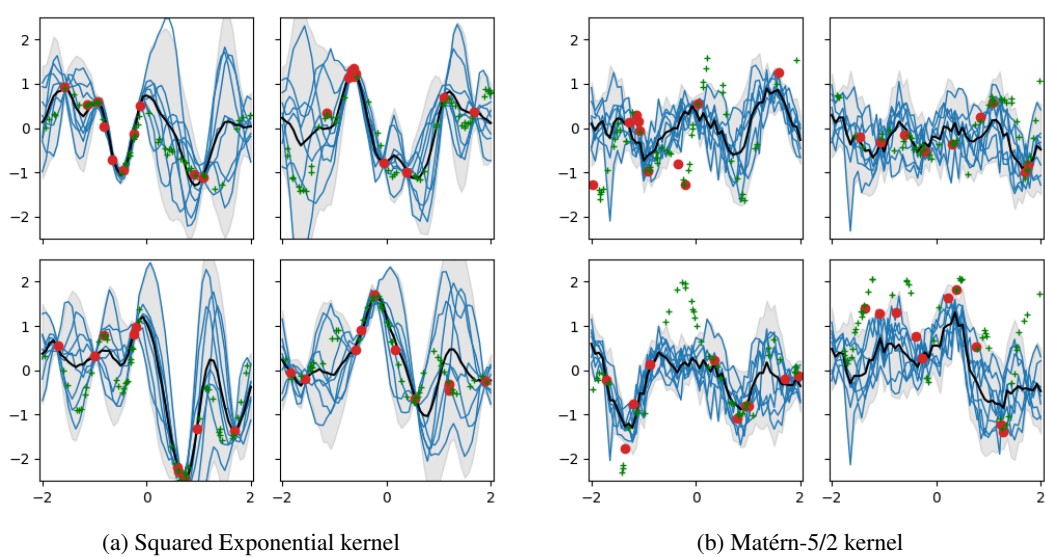

(a) Squared Exponential kernel                         (b) Matérn-5/2 kernel

Figure 4: Function samples generated by NBP under two different GP kernels. In each plot, the black solid line indicates the sample mean, blue lines are function samples, red circles represent the context points, and green crosses denote the target points.

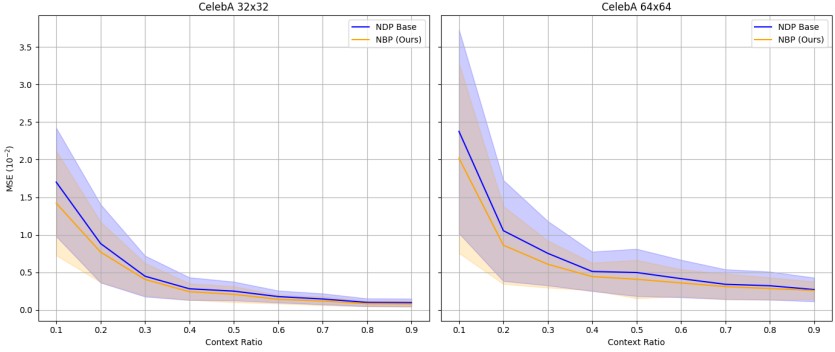

Figure 5: Comparison of reconstruction errors (MSE) between the NDP Base and our proposed NBP on the CelebA dataset at resolutions of 32×32 and 64×64. The horizontal axis represents the context ratio (i.e., the proportion of retained pixels), while the vertical axis shows the reconstruction error in units of $10^{-2}$. Solid lines indicate the mean MSE across test samples, and the shaded regions represent the standard deviation (Std), reflecting model uncertainty.

## B  Related Works

### B.1  Neural Processes and Their Extensions

Neural Processes (NPs) Garnelo et al. (2018b) combine the flexibility of neural networks with the uncertainty modeling capabilities of stochastic processes, aiming to learn distributions over functions. Conditional Neural Processes (CNPs) Garnelo et al. (2018a) extend this framework by conditioning on observed context points to predict target outputs. Attentive Neural Processes (ANPs) Kim et al. (2019) further enhance NPs by incorporating attention mechanisms, enabling the model to focus on relevant context points for each target prediction. Despite these advancements, challenges remain in capturing complex, multimodal distributions and ensuring consistency in posterior predictions. These approaches have been successfully extended to various domains, including sequential modeling Singh et al. (2019); Nguyen & Grover (2022); Bruinsma et al. (2023), convolutional architectures Gordon et al. (2019); Foong et al. (2020), graph-based models Hu et al. (2023), and probabilistic predictive models for large language models (LLMs) Requeima et al. (2024).

### B.2  Diffusion and Bridge Models in Generative Modeling

Denoising Diffusion Probabilistic Models (DDPMs) Ho et al. (2020) are powerful generative models that approximate complex data distributions by reversing a progressive noising process. Conditional Diffusion Models (CDMs) Choi et al. (2021); Zhang et al. (2023); Zhu et al. (2023) extend this framework by incorporating auxiliary information, enabling conditional generation. More recently, Denoising Diffusion Bridge Models (DDBMs) Zhou et al. (2023); Yue et al. (2023); Zheng et al. (2024); Li et al. (2023); Peluchetti (2023); He et al. (2024); Shi et al. (2023); Naderiparizi et al. (2025) have been proposed as a natural alternative. DDBMs introduce diffusion bridges—stochastic processes that interpolate between two paired distributions given as endpoints—making them well-suited for tasks such as image-to-image translation. However, existing DDBMs are primarily designed to model transformations in data space and may fall short in fully capturing the stochastic nature of functional mappings.

In this work, we avoid the complexity and computational overhead of SDE-based diffusion bridges. Instead, we extend the bridge concept within the DDPM framework Ho et al. (2020) through SNR-aware functional modeling. Since DDBMs address a different problem—focusing on generative modeling—while our work centers on functional learning, a direct experimental comparison with the original DDBM is not feasible.

### B.3  Generative Models for Function Modeling

Recent work has explored the use of diffusion models for function modeling. Neural Diffusion Processes (NDPs) Dutordoir et al. (2023) model distributions over functions by applying diffusion processes in latent space, allowing the representation of complex, non-Gaussian function distributions. Geometric Neural Diffusion Processes Mathieu et al. (2023) further extend this approach by incorporating geometric priors for infinite-dimensional modeling in non-Euclidean spaces. In parallel, other generative modeling techniques such as Neural ODEs Chen et al. (2018); Norcliffe et al. (2021), flow matching Lipman et al. (2022); Hamad & Rosenbaum, and score-based SDE methods Song et al. (2020); Dou et al. (2025) are also being integrated into the Neural Processes (NP) framework to enhance function modeling capabilities. In our experiments, we compared against open-source methods, including score-based neural processes (SNP) Dou et al. (2025) and Geometric Neural Diffusion Processes Mathieu et al. (2023), both of which demonstrated the empirical advantages of our approach.

## C  Formulation of the DBP Framework

### C.1  Derivation of Equation (10) in the Main Text

In our setting, the forward process also depends on $x$. Assume that $y_0$ is the initial state, which may correspond to $y_0$ or another variable. For clarity, we assume $y_0$ is the initial state and $x$ is the target.

From $y_{t-1}$ to $y_t$:

$$y_t = \sqrt{1 - \beta_t} y_{t-1} + \gamma_t x + \sqrt{\beta_t} \epsilon_t \tag{23}$$

By recursively expanding:

$$
\begin{aligned}
y_t &= \sqrt{1 - \beta_t} \left( \sqrt{1 - \beta_{t-1}} y_{t-2} + \gamma_{t-1} x + \sqrt{\beta_{t-1}} \epsilon_{t-1} \right) + \gamma_t x + \sqrt{\beta_t} \epsilon_t \\
&= \sqrt{(1 - \beta_t)(1 - \beta_{t-1})} y_{t-2} + \left( \sqrt{1 - \beta_t} \gamma_{t-1} + \gamma_t \right) x + \text{noise terms}
\end{aligned}
\tag{24}
$$

Continuing this expansion, we obtain:

$$y_t = \left( \prod_{s=1}^{t} \sqrt{1 - \beta_s} \right) y_0 + \left( \sum_{s=1}^{t} \gamma_s \prod_{k=s+1}^{t} \sqrt{1 - \beta_k} \right) x + \text{noise terms} \tag{25}$$

Define $\bar{\alpha}_t = \prod_{s=1}^{t}(1 - \beta_s)$, then,

$$y_t = \sqrt{\bar{\alpha}_t} y_0 + \left( \sum_{s=1}^{t} \gamma_s \sqrt{\frac{\bar{\alpha}_t}{\bar{\alpha}_s}} \right) x + \text{noise terms} \tag{26}$$

We define the cumulative bridge coefficient $\bar{\gamma}_t$ as

$$\bar{\gamma}_t = \sum_{s=1}^{t} \gamma_s \sqrt{\frac{\bar{\alpha}_t}{\bar{\alpha}_s}}. \tag{27}$$

The noise terms arise from the $\sqrt{\beta_s} \epsilon_s$ contributions at each step

$$
\begin{aligned}
\text{noise terms} =& \sqrt{\beta_t} \epsilon_t + \sqrt{1 - \beta_t} \sqrt{\beta_{t-1}} \epsilon_{t-1} + \sqrt{(1 - \beta_t)(1 - \beta_{t-1})} \sqrt{\beta_{t-2}} \epsilon_{t-2} \\
&+ \cdots + \sqrt{\bar{\alpha}_t / \bar{\alpha}_1} \sqrt{\beta_1} \epsilon_1.
\end{aligned}
\tag{28}
$$

This can be written compactly as:

$$\text{noise terms} = \sum_{s=1}^{t} \left( \sqrt{\beta_s} \prod_{k=s+1}^{t} \sqrt{1 - \beta_k} \right) \epsilon_s.$$

Since the $\epsilon_s$ are independent, the total variance is the sum of the variances of each term:

$$\text{Var(noise terms)} = \sum_{s=1}^{t} \beta_s \prod_{k=s+1}^{t} (1 - \beta_k).$$

we rewrite the variance:

$$\text{Var(noise terms)} = \sum_{s=1}^{t} \beta_s \frac{\bar{\alpha}_t}{\bar{\alpha}_s}.$$

Using $\bar{\alpha}_s = \prod_{k=1}^{s}(1 - \beta_k)$, we can express $\beta_s$ as $\beta_s = 1 - (1 - \beta_s) = 1 - \frac{\bar{\alpha}_s}{\bar{\alpha}_{s-1}}$. Substituting this in:

$$\text{Var(noise terms)} = \sum_{s=1}^{t} \left( 1 - \frac{\bar{\alpha}_s}{\bar{\alpha}_{s-1}} \right) \frac{\bar{\alpha}_t}{\bar{\alpha}_s}.$$

This simplifies to:

$$\text{Var(noise terms)} = \sum_{s=1}^{t} \left( \frac{\bar{\alpha}_t}{\bar{\alpha}_s} - \frac{\bar{\alpha}_t}{\bar{\alpha}_{s-1}} \right).$$

This is a telescoping series:

$$\text{Var(noise terms)} = \left( \frac{\bar{\alpha}_t}{\bar{\alpha}_1} - \frac{\bar{\alpha}_t}{\bar{\alpha}_0} \right) + \left( \frac{\bar{\alpha}_t}{\bar{\alpha}_2} - \frac{\bar{\alpha}_t}{\bar{\alpha}_1} \right) + \cdots + \left( \frac{\bar{\alpha}_t}{\bar{\alpha}_t} - \frac{\bar{\alpha}_t}{\bar{\alpha}_{t-1}} \right).$$

Most terms cancel out, leaving:

$$\text{Var(noise terms)} = \bar{\alpha}_t \left( \frac{1}{\bar{\alpha}_t} - \frac{1}{\bar{\alpha}_0} \right).$$

Assuming $\bar{\alpha}_0 = 1$ (since no steps have been applied at $t = 0$):

$$\text{Var(noise terms)} = 1 - \bar{\alpha}_t.$$

The variance of the accumulated noise can be computed similarly to the DDPM framework and is assumed to be: $(1 - \bar{\alpha}_t)I$. Thus, Equation (10) in the main text has been proven.

## C.2 DERIVATION OF EQUATION (16) AND (17) IN THE MAIN TEXT

### C.2.1 THE REVERSE PROCESS POSTERIOR

The marginal distribution $q(y_{t-1}|y_0, x)$ is:

$$q(y_{t-1}|y_0, x) = \mathcal{N}\left( \sqrt{\bar{\alpha}_{t-1}} y_0 + \bar{\gamma}_{t-1} x, (1 - \bar{\alpha}_{t-1})I \right), \tag{29}$$

where $\bar{\alpha}_t = \prod_{s=1}^{t}(1 - \beta_s)$ and $\bar{\gamma}_t = \sum_{s=1}^{t} \gamma_s \sqrt{\frac{\bar{\alpha}_t}{\bar{\alpha}_s}}$.

The reverse process posterior is:

$$q(y_{t-1}|y_t, y_0, x) \propto q(y_t|y_{t-1}, x)q(y_{t-1}|y_0, x). \tag{30}$$

This is a product of two Gaussians:

$$\mathcal{N}(y_t; Ay_{t-1} + b, \sigma_1^2 I) \times \mathcal{N}(y_{t-1}; \mu, \sigma_2^2 I), \tag{31}$$

where:

- $A = \sqrt{1 - \beta_t}$,
- $b = \gamma_t x$,
- $\sigma_1^2 = \beta_t$,
- $\mu = \sqrt{\bar{\alpha}_{t-1}} y_0 + \bar{\gamma}_{t-1} x$,
- $\sigma_2^2 = 1 - \bar{\alpha}_{t-1}$.

Assume

$$q(y_{t-1}|y_t, y_0, x) = \mathcal{N}(\tilde{\mu}, \tilde{\beta}_t I)$$

The mean of the product is:

$$\tilde{\mu} = \left( \frac{A^T(y_t - b)}{\sigma_1^2} + \frac{\mu}{\sigma_2^2} \right) \left( \frac{A^T A}{\sigma_1^2} + \frac{1}{\sigma_2^2} \right)^{-1}. \tag{32}$$

Substituting the values:

$$\tilde{\mu} = \left( \frac{\sqrt{1 - \beta_t}(y_t - \gamma_t x)}{\beta_t} + \frac{\sqrt{\bar{\alpha}_{t-1}} y_0 + \bar{\gamma}_{t-1} x}{1 - \bar{\alpha}_{t-1}} \right) \left( \frac{1 - \beta_t}{\beta_t} + \frac{1}{1 - \bar{\alpha}_{t-1}} \right)^{-1}. \tag{33}$$

Simplify the denominator:

$$\frac{1 - \beta_t}{\beta_t} + \frac{1}{1 - \bar{\alpha}_{t-1}} = \frac{(1 - \beta_t)(1 - \bar{\alpha}_{t-1}) + \beta_t}{\beta_t(1 - \bar{\alpha}_{t-1})} = \frac{1 - \bar{\alpha}_t}{\beta_t(1 - \bar{\alpha}_{t-1})}, \tag{34}$$

since $\bar{\alpha}_t = (1 - \beta_t)\bar{\alpha}_{t-1}$.

Thus:

$$\tilde{\mu} = \left( \frac{\sqrt{1 - \beta_t}(y_t - \gamma_t x)}{\beta_t} + \frac{\sqrt{\bar{\alpha}_{t-1}} y_0 + \bar{\gamma}_{t-1} x}{1 - \bar{\alpha}_{t-1}} \right) \frac{\beta_t(1 - \bar{\alpha}_{t-1})}{1 - \bar{\alpha}_t}. \tag{35}$$

Now, expand the numerator:

$$\tilde{\mu} = \frac{\sqrt{1 - \beta_t}(1 - \bar{\alpha}_{t-1})}{1 - \bar{\alpha}_t}(y_t - \gamma_t x) + \frac{\beta_t(\sqrt{\bar{\alpha}_{t-1}}y_0 + \bar{\gamma}_{t-1}x)}{1 - \bar{\alpha}_t}. \tag{36}$$

This can be rewritten as:

$$\tilde{\mu} = \frac{\sqrt{1 - \beta_t}(1 - \bar{\alpha}_{t-1})}{1 - \bar{\alpha}_t}y_t + \frac{\beta_t\sqrt{\bar{\alpha}_{t-1}}}{1 - \bar{\alpha}_t}y_0 + \left(\frac{\beta_t\bar{\gamma}_{t-1}}{1 - \bar{\alpha}_t} - \frac{\sqrt{1 - \beta_t}(1 - \bar{\alpha}_{t-1})\gamma_t}{1 - \bar{\alpha}_t}\right)x. \tag{37}$$

The variance $\tilde{\beta}_t$ is derived as:

$$\tilde{\beta}_t = \left(\frac{1 - \beta_t}{\beta_t} + \frac{1}{1 - \bar{\alpha}_{t-1}}\right)^{-1} = \frac{\beta_t(1 - \bar{\alpha}_{t-1})}{1 - \bar{\alpha}_t}. \tag{38}$$

### C.2.2 REPARAMETERIZATION OF $y_0$ (SIMILAR TO DDPM)

From the forward process, we can express $y_t$ as:

$$y_t = \sqrt{\bar{\alpha}_t}y_0 + \bar{\gamma}_t x + \sqrt{1 - \bar{\alpha}_t}\epsilon, \tag{39}$$

where $\epsilon \sim \mathcal{N}(0, I)$. Solving for $y_0$:

$$y_0 = \frac{y_t - \bar{\gamma}_t x - \sqrt{1 - \bar{\alpha}_t}\epsilon}{\sqrt{\bar{\alpha}_t}}. \tag{40}$$

Substitute $y_0$ into the Reverse Process Mean $\tilde{\mu}$

The derived mean $\tilde{\mu}$ is:

$$\tilde{\mu} = \frac{\sqrt{1 - \beta_t}(1 - \bar{\alpha}_{t-1})y_t + \beta_t\sqrt{\bar{\alpha}_{t-1}}y_0 + (\beta_t\bar{\gamma}_{t-1} - \sqrt{1 - \beta_t}(1 - \bar{\alpha}_{t-1})\gamma_t)x}{1 - \bar{\alpha}_t}. \tag{41}$$

Substitute $y_0$:

$$\tilde{\mu} = \frac{\sqrt{1 - \beta_t}(1 - \bar{\alpha}_{t-1})y_t + \beta_t\sqrt{\bar{\alpha}_{t-1}}\left(\frac{y_t - \bar{\gamma}_t x - \sqrt{1 - \bar{\alpha}_t}\epsilon}{\sqrt{\bar{\alpha}_t}}\right) + (\beta_t\bar{\gamma}_{t-1} - \sqrt{1 - \beta_t}(1 - \bar{\alpha}_{t-1})\gamma_t)x}{1 - \bar{\alpha}_t}. \tag{42}$$

Simplify the Expression:

1. Combine $y_t$ Terms:

$$\frac{\sqrt{1 - \beta_t}(1 - \bar{\alpha}_{t-1})y_t + \frac{\beta_t\sqrt{\bar{\alpha}_{t-1}}y_t}{\sqrt{\bar{\alpha}_t}}}{1 - \bar{\alpha}_t} = \frac{\left(\sqrt{1 - \beta_t}(1 - \bar{\alpha}_{t-1}) + \frac{\beta_t}{\sqrt{1 - \beta_t}}\right)y_t}{1 - \bar{\alpha}_t},$$
$$= \frac{y_t}{\sqrt{1 - \beta_t}} \tag{43}$$

where we used $\bar{\alpha}_t = \bar{\alpha}_{t-1}(1 - \beta_t)$, so $\sqrt{\bar{\alpha}_{t-1}}/\sqrt{\bar{\alpha}_t} = 1/\sqrt{1 - \beta_t}$.

2. Combine $x$ Terms:

$$\frac{-\frac{\beta_t\sqrt{\bar{\alpha}_{t-1}}\bar{\gamma}_t}{\sqrt{\bar{\alpha}_t}} + \beta_t\bar{\gamma}_{t-1} - \sqrt{1 - \beta_t}(1 - \bar{\alpha}_{t-1})\gamma_t}{1 - \bar{\alpha}_t}x \triangleq C(t)x \tag{44}$$

3. Noise ($\epsilon$) Term:

$$-\frac{\beta_t\sqrt{\bar{\alpha}_{t-1}}\sqrt{1 - \bar{\alpha}_t}\epsilon}{\sqrt{\bar{\alpha}_t}(1 - \bar{\alpha}_t)} = -\frac{\beta_t}{\sqrt{1 - \beta_t}\sqrt{1 - \bar{\alpha}_t}}\epsilon. \tag{45}$$

Then, the mean can be written as:

$$\tilde{\mu} = \frac{1}{\sqrt{1 - \beta_t}}\left(y_t - \frac{\beta_t}{\sqrt{1 - \bar{\alpha}_t}}\epsilon\right) + C(t)x. \tag{46}$$

which proves the Equation (16) in the main text.

### C.2.3 CALCULATION OF $C(t)$

From Equation (44),

$$C(t) = \frac{-\frac{\beta_t \sqrt{\bar{\alpha}_{t-1}} \bar{\gamma}_t}{\sqrt{\bar{\alpha}_t}} + \beta_t \bar{\gamma}_{t-1} - \sqrt{1 - \beta_t}(1 - \bar{\alpha}_{t-1})\gamma_t}{1 - \bar{\alpha}_t}. \tag{47}$$

Using $\sqrt{\bar{\alpha}_t} = \sqrt{\bar{\alpha}_{t-1}}\sqrt{1 - \beta_t}$, we first compute:

$$-\frac{\beta_t \sqrt{\bar{\alpha}_{t-1}} \bar{\gamma}_t}{\sqrt{\bar{\alpha}_{t-1}}\sqrt{1 - \beta_t}} + \beta_t \bar{\gamma}_{t-1} = -\frac{\beta_t \bar{\gamma}_t}{\sqrt{1 - \beta_t}} + \beta_t \bar{\gamma}_{t-1}. \tag{48}$$

Next, using the relation $\bar{\gamma}_t = \gamma_t + \sqrt{1 - \beta_t}\bar{\gamma}_{t-1}$ (since $\bar{\gamma}_t = \sum_{s=1}^{t} \gamma_s \sqrt{\frac{\bar{\alpha}_t}{\bar{\alpha}_s}} = \gamma_t + \sqrt{\frac{\bar{\alpha}_t}{\bar{\alpha}_{t-1}}}\bar{\gamma}_{t-1} = \gamma_t + \sqrt{1 - \beta_t}\bar{\gamma}_{t-1}$), we simplify:

$$-\frac{\beta_t \bar{\gamma}_t}{\sqrt{1 - \beta_t}} + \beta_t \bar{\gamma}_{t-1} = -\frac{\beta_t(\gamma_t + \sqrt{1 - \beta_t}\bar{\gamma}_{t-1})}{\sqrt{1 - \beta_t}} + \beta_t \bar{\gamma}_{t-1} = -\frac{\beta_t \gamma_t}{\sqrt{1 - \beta_t}} - \beta_t \bar{\gamma}_{t-1} + \beta_t \bar{\gamma}_{t-1}$$

$$= -\frac{\beta_t \gamma_t}{\sqrt{1 - \beta_t}}. \tag{49}$$

Thus, the final bridge correction term is:

$$C_t(x) = \frac{-\frac{\beta_t \gamma_t}{\sqrt{1 - \beta_t}} - \sqrt{1 - \beta_t}(1 - \bar{\alpha}_{t-1})\gamma_t}{1 - \bar{\alpha}_t}x = -\frac{\beta_t + (1 - \beta_t)(1 - \bar{\alpha}_{t-1})}{\sqrt{1 - \beta_t}(1 - \bar{\alpha}_t)}\gamma_t x. \tag{50}$$

Using $\sqrt{\bar{\alpha}_t} = \sqrt{\bar{\alpha}_{t-1}}\sqrt{1 - \beta_t}$, we obtain the simplified form:

$$C(t)x = -\frac{\gamma_t}{\sqrt{1 - \beta_t}}x. \tag{51}$$

Substitute $\gamma_t = \frac{\text{SNR}_T}{\text{SNR}_t} = \frac{\bar{\alpha}_T(1 - \bar{\alpha}_t)}{\bar{\alpha}_t(1 - \bar{\alpha}_T)}$:

$$C(t) = -\frac{1}{\sqrt{1 - \beta_t}} \cdot \frac{\bar{\alpha}_T(1 - \bar{\alpha}_t)}{\bar{\alpha}_t(1 - \bar{\alpha}_T)}. \tag{52}$$

In essence, the role of $C(t)$ is to correct the contribution of $x$ during the reverse process, ensuring that the generative procedure properly incorporates the bridging information.

### C.3 DERIVATION OF EQUATION (18) IN THE MAIN TEXT

We derive the NBP loss $\mathcal{L}_\theta = \mathbb{E}_{t, y_0, x, \epsilon}\left[\|\epsilon_\theta(y_t, x, t) - \epsilon\|_2^2\right]$ in Equation (18) in the main text directly from the Evidence Lower Bound (ELBO).

The log-likelihood of the data $y_0$ is lower-bounded by:

$$\log p_\theta(y_0|x) \geq \mathbb{E}_{q(y_{1:T}|y_0, x)}\left[\log \frac{p_\theta(y_{0:T}|x)}{q(y_{1:T}|y_0, x)}\right] = \text{ELBO}, \tag{53}$$

where:

• $p_\theta(y_{0:T}|x)$ is the reverse (generative) process.

• $q(y_{1:T}|y_0, x)$ is the forward (noising) process.

1. The ELBO decomposes into:

$$\text{ELBO} = \mathbb{E}_{q(y_{1:T}|y_0,x)} \left[ \log p(y_T|x) + \sum_{t=2}^{T} \log \frac{p_\theta(y_{t-1}|y_t,x)}{q(y_t|y_{t-1},x)} + \log \frac{p_\theta(y_0|y_1,x)}{q(y_1|y_0,x)} \right]$$

$$= \mathbb{E}_{q(y_{1:T}|y_0,x)} \left[ \log \frac{p(y_T|x)}{q(y_T|y_0,x)} + \sum_{t=2}^{T} \log \frac{p_\theta(y_{t-1}|y_t,x)}{q(y_{t-1}|y_t,y_0,x)} + \log p_\theta(y_0|y_1,x) \right] \tag{54}$$

The key term is the sum of KL divergences between $p_\theta(y_{t-1}|y_t,x)$ and $q(y_{t-1}|y_t,y_0,x)$:

$$\mathbb{E}_{q(y_{1:T}|y_0,x)} \left[ \sum_{t=2}^{T} D_{\text{KL}}(q(y_{t-1}|y_t,y_0,x)\|p_\theta(y_{t-1}|y_t,x)) \right] \tag{55}$$

The KL divergence term between $q(y_{t-1}|y_t,y_0,x)$ and $p_\theta(y_{t-1}|y_t,x)$ is:

$$D_{\text{KL}}(q\|p_\theta) \propto \|\mu_\theta(y_t,t,x) - \tilde{\mu}(y_t,y_0,x)\|^2. \tag{56}$$

where $\mu_\theta(y_t,t,x) - \tilde{\mu}(y_t,y_0,x)$ are the means of $q(y_{t-1}|y_t,y_0,x)$ and $p_\theta(y_{t-1}|y_t,x)$, respectively.

2. Sample from $q(y_t|y_0,x)$: Using the forward process properties by Equation (10) in the main text, we can write:

$$y_t = \sqrt{\bar{\alpha}_t}y_0 + \bar{\gamma}_t x + \sqrt{1-\bar{\alpha}_t}\epsilon, \quad \epsilon \sim \mathcal{N}(0, \mathbf{I}) \tag{57}$$

This allows sampling $y_t$ directly from $y_0$, $x$ and $\epsilon$.

3. Rewrite $q(y_{t-1}|y_t,y_0,x)$ by Equation (46):

$$\tilde{\mu}_t(y_t,y_0,x) = \frac{1}{\sqrt{1-\beta_t}} \left( y_t - \frac{\beta_t}{\sqrt{1-\bar{\alpha}_t}}\epsilon \right) + C(t)x. \tag{58}$$

4. Reparameterize $\mu_\theta(y_t,t,x)$: Assume $p_\theta(y_{t-1}|y_t,x)$ predicts $\tilde{\mu}_t$:

$$\mu_\theta(y_t,t,x) = \frac{1}{\sqrt{1-\beta_t}} \left( y_t - \frac{\beta_t}{\sqrt{1-\bar{\alpha}_t}}\epsilon_\theta(y_t,t,x) \right) + C(t)x \tag{59}$$

Here, $\epsilon_\theta(y_t,t,x)$ is a neural network predicting the noise $\epsilon$.

5. Final Noise-Prediction Loss: The KL terms simplify to a weighted $L_2$ loss on the noise:

$$\mathbb{E}_{t,\epsilon} \left[ \frac{\beta_t^2}{2\sigma_t^2\alpha_t(1-\bar{\alpha}_t)} \|\epsilon - \epsilon_\theta(y_t,t,x)\|^2 \right] \tag{60}$$

Dropping the weighting (as in DDPM) gives the simplified loss:

$$\mathcal{L} = \mathbb{E}_{t,y_0,x,\epsilon} \left[ \|\epsilon - \epsilon_\theta(y_t,t,x)\|^2 \right], \tag{61}$$

where:

• $t \sim \text{Uniform}(1,T)$,

• $y_t = \sqrt{\bar{\alpha}_t}y_0 + \bar{\gamma}_t x + \sqrt{1-\bar{\alpha}_t}\epsilon$.

This proves Equation (18) in the main text, and the denoising network $\epsilon_\theta(y_t,t,x)$ is self-consistent with respect to the condition on $x$.

## D  BACKGROUND: NEURAL PROCESSES

Neural Processes (NPs) Garnelo et al. (2018b;a); Kim et al. (2019); Louizos et al. (2019) combine the expressiveness of neural networks with the probabilistic reasoning of Gaussian Processes (GPs) Rasmussen (2003). While GPs offer principled uncertainty quantification, they suffer from poor scalability Snelson & Ghahramani (2005); Titsias (2009) and limited kernel flexibility Wilson et al. (2016); Liu et al. (2021). In contrast, Neural Networks (NNs) Schmidhuber (2015); Nielsen (2015) are highly flexible and scalable but lack inherent mechanisms for uncertainty modeling Blundell et al.

(2015); Pearce et al. (2020); Gawlikowski et al. (2023). NPs address these limitations by modeling distributions over functions using a neural network-based framework. They approximate a stochastic process $F : X \to Y$ through finite-dimensional marginals, parameterized by a latent variable $z$ to capture global uncertainty. Given context observations $(x_\mathbb{C}, y_\mathbb{C})$ and target inputs $x_\mathbb{T}$, NPs generate predictive distributions over $y_\mathbb{T}$ via a conditional latent model.

$$p(y_\mathbb{T}, z | x_\mathbb{T}, x_\mathbb{C}, y_\mathbb{C}) = p(z | x_\mathbb{C}, y_\mathbb{C}) \prod_{i=1}^{|\mathbb{T}|} p(y_{\mathbb{T},i} | x_{\mathbb{T},i}, z) \tag{62}$$

To ensure computational efficiency and order-invariance, NPs introduce three main components:
1. Encoder $h$: Maps each context input-output pair $(x_i, y_i)$ to a representation space, producing representations $r_i = h(x_i, y_i)$.

2. Aggregator $a$: Combines the encoded inputs into a single, permutation-invariant global representation $r$. This is typically done by averaging the representations: $r = a(r_i) = \frac{1}{|\mathbb{C}|} \sum_{i \in \mathbb{C}} r_i$. This global representation $r$ parameterizes the latent distribution $z \sim \mathcal{N}(\mu(r), I\sigma(r))$.

3. Decoder $g$: Predicts the target outputs $y_\mathbb{T} = g(x_\mathbb{T}, z)$, conditioned on the latent variable $z$ and the target inputs $x_\mathbb{T}$.

Here, $z$ encodes the uncertainty about the global structure of the underlying function. Training NPs uses amortized variational inference, optimizing an evidence lower bound (ELBO) on the conditional log-likelihood:

$$\log p(y_\mathbb{T} | x_\mathbb{C}, y_\mathbb{C}, x_\mathbb{T}) \geq \mathbb{E}_{q(z|x_\mathbb{C}, y_\mathbb{C})} \left[ \sum_{i \in \mathbb{T}} \log p(y_{\mathbb{T},i} | z, x_{\mathbb{T},i}) + \log \frac{p(z | x_\mathbb{C}, y_\mathbb{C})}{q(z | x_\mathbb{C}, y_\mathbb{C})} \right] \tag{63}$$

where $q(z | x_\mathbb{C}, y_\mathbb{C})$ is the variational posterior distribution parameterized by a neural network, and $p(z | x_\mathbb{C}, y_\mathbb{C})$ is the conditional prior.

# E NDP REVIEW

Formally, given a function $f : \mathbb{R}^D \to \mathbb{R}$, an NDP learns a generative distribution over observed data pairs $(x, y)$, where inputs $x \in \mathbb{R}^{N \times D}$ and outputs $y = f(x) \in \mathbb{R}^N$. Unlike standard NPs, NDPs Dutordoir et al. (2023) do not explicitly require a partitioning into context and target sets during training; all points are jointly modeled. In supervised learning setting, the NDP modeling framework consists of two stochastic processes:

**Forward Diffusion Process.** Starting from observed clean data $y_0$, the forward diffusion process gradually injects Gaussian noise into the outputs over $T$ timesteps according to a predefined variance schedule $\{\beta_t\}$:

$$q(y_{1:T} | y_0) = \prod_{t=1}^{T} q(y_t | y_{t-1}), \quad q(y_t | y_{t-1}) = \mathcal{N}(y_t; \sqrt{1 - \beta_t}\, y_{t-1}, \beta_t I). \tag{64}$$

After $T$ diffusion steps, the distribution of the outputs converges towards standard Gaussian noise, i.e., $y_T \sim \mathcal{N}(0, I)$.

**Reverse Process.** Neural Diffusion Processes (NDPs) learn a conditional reverse process that denoises observations from Gaussian noise $y_T$ to outputs $y_0$, guided by an input $x$:

$$p_\theta(y_{0:T} | x) = p(y_T) \prod_{t=1}^{T} p_\theta(y_{t-1} | y_t, x), \tag{65}$$

with Gaussian transitions parameterized by a noise prediction model $\epsilon_\theta$:

$$p_\theta(y_{t-1} | y_t, x) = \mathcal{N}\left(y_{t-1}; \mu_\theta(y_t, t, x), \tilde{\beta}_t I\right). \tag{66}$$

where $\tilde{\beta}_t = \frac{1-\bar{\alpha}_{t-1}}{1-\bar{\alpha}_t}\beta_t$, $\mu_\theta(y_t, t, x)$ is reparameterized as

$$\mu_\theta(y_t, t, x) = \frac{1}{\sqrt{1-\beta_t}}\left(y_t - \frac{\beta_t}{\sqrt{1-\bar{\alpha}_t}}\epsilon_\theta(y_t, t, x)\right), \quad \bar{\alpha}_t = \prod_{s=1}^{t}(1-\beta_s) \tag{67}$$

**Training Objective.** NDPs employ a denoising score matching objective Hyvärinen & Dayan (2005); Song et al. (2021); Huang et al. (2021), training the noise model $\epsilon_\theta$ by minimizing the discrepancy between predicted noise and actual noise $\epsilon \sim \mathcal{N}(0, I)$:

$$\mathcal{L}_\theta = \mathbb{E}_{t,x,y_0,\epsilon}\left[\|\epsilon - \epsilon_\theta(y_t, t, x)\|_2^2\right], \quad \text{with} \quad y_t = \sqrt{\bar{\alpha}_t}y_0 + \sqrt{1-\bar{\alpha}_t}\epsilon. \tag{68}$$

**Conditional Sampling Procedure.** At test time, NDPs draw samples from the conditional distribution $p(y_{\mathbb{T},0} \mid x_{\mathbb{T}}, D)$, where $D = (x_{\mathbb{C}} \in \mathbb{R}^{M \times D}, y_{\mathbb{C},0} \in \mathbb{R}^M)$ is the context observations.

The conditional sampling proceeds as follows. First, sample the initial target noise: $y_{\mathbb{T},T} \sim \mathcal{N}(0, I)$. For each diffusion timestep $t = T, \ldots, 1$, proceed with:

- Sample the noisy version of the context points using the forward diffusion process:

$$y_{\mathbb{C},t} \sim \mathcal{N}\left(\sqrt{\bar{\alpha}_t}y_{\mathbb{C},0}, (1-\bar{\alpha}_t)I\right), \tag{69}$$

- Form the combined dataset at time $t$ by collecting the union of noisy targets and noisy contexts:

$$y_t = \{y_{\mathbb{T},t}, y_{\mathbb{C},t}\}, \quad x = \{x_{\mathbb{T}}, x_{\mathbb{C}}\}. \tag{70}$$

- Perform the reverse denoising step by sampling from the learned backward kernel:

$$y_{t-1} \sim \mathcal{N}\left(\frac{1}{\sqrt{\alpha_t}}\left(y_t - \frac{\beta_t}{\sqrt{1-\bar{\alpha}_t}}\epsilon_\theta(y_t, t, x)\right), \sigma_t^2 I\right), \quad \text{where} \quad \alpha_t = 1 - \beta_t. \tag{71}$$

| Method | Endpoint Match | Path Consistency |
|---|---|---|
| NDP (Baseline) | Implicit | Weak |
| NBP (Ours) | ✓ | ✓ |

Table 3: Comparison of generation properties.

# F  NOISE MODEL ARCHITECTURE

To ensure that our model remains consistent with the structural properties of stochastic processes and to guarantee fair experimental comparisons, we adopt the same noise model architecture as NDPs, namely the Bi-Dimensional Attention Block Dutordoir et al. (2023).

As decsribed in Figure 1, this architecture is designed to encode two key symmetries:

- **Exchangeability over data points:** the model should be equivariant to permutations of the dataset ordering. That is, shuffling the order of inputs in the context or target set should not affect the output distribution.

- **Invariance over input dimensions:** the prediction should be unaffected by reordering of input features (e.g., swapping the order of columns in a tabular dataset).

To accommodate both properties, the Bi-Dimensional Attention Block operates over a tensor $s_t \in \mathbb{R}^{N \times D \times H}$ representing the latent representation of paired inputs $(x, y_t)$ and timestep $t$. Each block consists of two multi-head self-attention (MHSA) mechanisms:

- MHSA$_N$: acts across the *dataset axis* $N$, propagating information across data points;
- MHSA$_D$: acts across the *input dimension axis* $D$, capturing interactions between features.

The output of each block at layer $\ell$ is computed as:

$$A_t^{(\ell)}(s_t^{(\ell-1)}) = A_t^{(\ell-1)} + \sigma\left(\text{MHSA}_N(s_t^{(\ell-1)}) + \text{MHSA}_D(s_t^{(\ell-1)})\right),$$

where $\sigma$ denotes a ReLU activation, and $A_t^{(0)} = 0$, $s_t^{(0)} = s_t$ is the output of the preprocessing stage.

Each Bi-Dimensional Attention Block maintains equivariance under permutations of data and feature dimensions:

**Proposition 1 (Equivariance (Dutordoir et al., 2023, Prop. 4.1))** *Let $\pi_N$ and $\pi_D$ be permutations over dataset and feature axes respectively. Then,*

$$\pi_D \circ \pi_N \circ A_t(s) = A_t(\pi_D \circ \pi_N \circ s), \quad \forall s \in \mathbb{R}^{N \times D \times H}.$$

The final noise model $\epsilon_\theta$ is obtained by summing outputs across all Bi-Attention layers, followed by an aggregation over the input dimension axis to remove dependence on feature order. This leads to:

**Proposition 2 ( Equivariance (Dutordoir et al., 2023, Prop. 4.2))** *Let $\pi_N, \pi_D$ be permutations as above. Then,*

$$\pi_N \circ \epsilon_\theta(x_t, y_t, t) = \epsilon_\theta(\pi_N \circ \pi_D \circ x_t, \pi_N \circ y_t, t).$$

By directly encoding these properties into the noise model architecture, NDPs and NBPs ensure that the predicted outputs $\{y_t^1, \ldots, y_t^N\}$ at each timestep $t$ form an *exchangeable* set of random variables, consistent with the Kolmogorov Extension Theorem (KET). This is critical for defining a valid stochastic process over functions.

## G  MORE DETAILS FOR EXPERIMENTS

### G.1  BASELINE IMPLEMENTATION AND EVALUATION METRICS

To provide a comprehensive comparison, we implement NPs Garnelo et al. (2018b) , ANPs Kim et al. (2019), and ConvNPs Gordon et al. (2019) using the official NP-Family repository Dubois et al. (2020), with all hyperparameters set to the recommended default values. For NDP Dutordoir et al. (2023), we directly utilize the official repositories. Meanwhile, to ensure a fair comparison, our NBP model adopts the same Bi-Dimensional Attention Block architecture and hyperparameters as the baseline NDP. The implementation details are provided in the supplementary materials. All models are retrained on the experimental datasets to ensure consistent metric evaluation and visualization. We evaluate model performance using two primary metrics: Mean Squared Error (MSE) and Negative Log-Likelihood (NLL). All experiments are conducted on a single NVIDIA RTX 4090 GPU.

### G.2  DETAILS OF EEG DATASET REGRESSION TASK

The EEG dataset used in this experiment consists of recordings from 122 subjects, including both alcoholic and control groups. Each subject underwent either single or double stimulus conditions, during which neural responses were recorded using 64 scalp electrodes. Each trial lasted for 1 second with a sampling rate of 256 Hz, and up to 120 trials were recorded per subject.

For our study, we focus on signals from 7 frontal electrodes: FZ, F1, F2, F3, F4, F5, and F6. This selection yields a total of 7,632 multivariate time series, each comprising 256 time steps across 7 channels. These signals exhibit strong temporal dynamics and inter-channel correlations, making the dataset well-suited for evaluating the generalization and modeling capabilities of multi-output meta-learning models. The data is publicly available from the UCI Machine Learning Repository, with collection details described in Zhang et al. (1995). Figure 6 illustrates the signals recorded from these seven channels for a single trial of one subject.

Subjects were split into training, validation, and test sets on a per-individual basis. The validation and test sets each contain 10 subjects, with the remainder assigned to the training set. All trials from each subject form a single meta-task, enabling task-level generalization evaluation.

Within each trial, we randomly select 3 out of the 7 channels and mask partial segments of these channels to simulate missing data. This setup supports evaluation across the following three tasks:

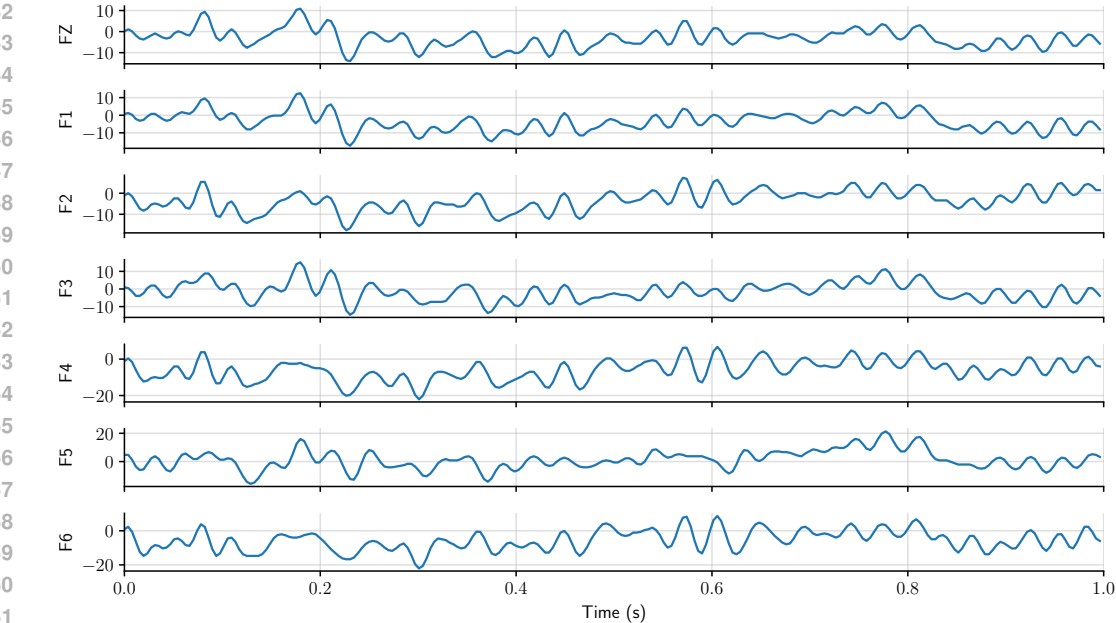

Figure 6: the signals recorded from these seven channels for a single trial of one subject.

**Interpolation:** Recovering locally missing values within the observed time range.

**Reconstruction:** Reconstructing entire masked regions of a target channel using the remaining channels as context.

**Forecasting:** Extrapolating future signal trajectories based on current observations.

Each input is represented as an index vector $\mathbf{x}_e = (i_t, i_c)$, where $i_t$ denotes the time step and $i_c$ the channel index. The corresponding output is the voltage signal $\mathbf{y}_e$.

All models were trained for 1,000 iterations on the training set. Evaluation metrics include Mean Squared Error (MSE) and Negative Log-Likelihood (NLL). The Neural Bridge Process (NBP) employs a 5-layer Bi-Dimensional Attention Block with hidden dimension 64 and 8 attention heads. The model was trained using a fixed random seed of 42 to ensure reproducibility.

For models incorporating diffusion-based generation (such as DNP and NBP), we adopt a cosine noise schedule with the following parameters: $\beta_{\text{start}} = 0.0003$, $\beta_{\text{end}} = 0.5$, and 500 diffusion timesteps. These settings are applied consistently across the forward and reverse processes in all diffusion-based models. Additional training hyperparameters are as follows:

- For NBP and DNP base, the learning rate was set to $2 \cdot 10^{-5}$;
- Other models used default learning rates as recommended in prior literature;
- All models operated on input sequences of 256 time steps.

The evaluation results, summarized in Table 1 of the main text, demonstrate that NBP consistently outperforms existing baseline models across all three tasks, highlighting its superior modeling capacity for highly correlated multichannel temporal data.

## G.3 Details of Image Regression Task

We provide detailed information on the image regression task using Neural Bridge Processes (NBPs). The task involves predicting pixel intensities based on their spatial coordinates, which are normalized to the range $[-2, 2]$. We use the CelebA dataset at resolutions of $32 \times 32$ and $64 \times 64$.

Our experimental protocol—including the denoising network architecture, training schedule, optimizer configuration, and random seed—closely follows the setup used for Neural Diffusion Processes

Table 4: CelebA $32 \times 32$ Results for NDP Base and NBP (Ours) in $10^{-2}$ MSE Units

| Context Ratio | Retained Pixels | NDP Base | | NBP (Ours) | |
|---|---|---|---|---|---|
| | | MSE Mean | MSE Std | MSE Mean | MSE Std |
| 0.1 | 96 | 1.7011 | 0.7221 | **1.4206** | 0.6958 |
| 0.2 | 197 | 0.8831 | 0.5214 | **0.7694** | 0.4008 |
| 0.3 | 312 | 0.4482 | 0.2724 | **0.4055** | 0.2157 |
| 0.4 | 412 | 0.2809 | 0.1511 | **0.2416** | 0.1091 |
| 0.5 | 512 | 0.2496 | 0.1251 | **0.2066** | 0.1062 |
| 0.6 | 570 | 0.1769 | 0.0793 | **0.1406** | 0.0573 |
| 0.7 | 714 | 0.1450 | 0.0729 | **0.1185** | 0.0579 |
| 0.8 | 832 | 0.0997 | 0.0518 | **0.0821** | 0.0373 |
| 0.9 | 913 | 0.0969 | 0.0527 | **0.0793** | 0.0392 |

Table 5: CelebA $64 \times 64$ Results for NDP Base and NBP (Ours) in $10^{-2}$ MSE Units

| Context Ratio | Retained Pixels | NDP Base | | NBP (Ours) | |
|---|---|---|---|---|---|
| | | MSE Mean | MSE Std | MSE Mean | MSE Std |
| 0.1 | 432 | 2.3763 | 1.3563 | **2.0189** | 1.2632 |
| 0.2 | 872 | 1.0546 | 0.6718 | **0.8615** | 0.5174 |
| 0.3 | 1276 | 0.7530 | 0.4283 | **0.6078** | 0.3132 |
| 0.4 | 1628 | 0.5115 | 0.2626 | **0.4432** | 0.1826 |
| 0.5 | 2080 | 0.4973 | 0.3144 | **0.4078** | 0.2548 |
| 0.6 | 2392 | 0.4160 | 0.2496 | **0.3587** | 0.1801 |
| 0.7 | 2788 | 0.3405 | 0.1985 | **0.3097** | 0.1694 |
| 0.8 | 3100 | 0.3215 | 0.1875 | **0.2838** | 0.1461 |
| 0.9 | 3492 | 0.2706 | 0.1579 | **0.2584** | 0.1177 |

(NDPs), ensuring a fair and consistent comparison. The core architecture of the NBP model consists of 7 layers, each with hidden dimension 64 and 8 attention heads. Sparse attention is not used in these experiments.

**Training Configuration.** The model is trained for 10 epochs using a batch size of 32. The optimizer follows a warmup and decay schedule:

- Initial learning rate: $2.0 \times 10^{-5}$

- Peak learning rate: $1.0 \times 10^{-3}$

- End learning rate: $1.0 \times 10^{-5}$

- Warmup: 20 epochs, decay over 200 epochs

- EMA decay rate: 0.995

**Diffusion Settings.** We employ a cosine beta schedule with the following parameters for the forward and reverse processes:

- $\beta_{\text{start}} = 0.0003$

- $\beta_{\text{end}} = 0.5$

- Number of timesteps: 500

**Evaluation Protocol.** Each prediction is averaged over 9 conditional samples during testing. The evaluation batch size is set to 9, with 128 samples drawn per image for final averaging. All pixel values are normalized to the $[0, 1]$ range. The model was trained and evaluated with a fixed random seed of 42.

**Loss Function.** We adopt the $\ell_1$ loss for training the denoising objective.

**Results and Analysis.** Tables 2 and 3 in the main text report the quantitative performance under various levels of context sparsity. NBPs consistently outperform NDPs across settings. For example, at a context ratio of 0.02 on CelebA $32 \times 32$, NBPs achieve an MSE of 0.76 compared to 0.88 by

NDPs. This trend persists at higher resolutions: on CelebA $64 \times 64$, NBPs achieve an MSE of 0.80 compared to NDPs' 1.05 under the same sparse context condition.

This performance gain is attributed to NBPs' novel design, wherein the forward diffusion kernel is explicitly conditioned on the input coordinates. This conditioning acts as a structural constraint across the diffusion trajectory, ensuring that the trajectory remains anchored to the input while steering toward the supervised target.

## H  CODE CONTRIBUTION

The full implementation of the Neural Bridge Processes (NBP) framework is provided in the supplementary materials to ensure reproducibility and to facilitate further evaluation by reviewers.

## I  STATEMENT ON THE USE OF LARGE LANGUAGE MODELS

Large language models (LLMs) were used solely for polishing and editing the text of this manuscript.

## J  LIMITATIONS

The current evaluation focuses exclusively on EEG and image regression tasks. Future work will explore the applicability of the proposed method to a broader range of domains, including spatiotemporal modeling, control, and scientific data analysis.

