# OpenReview forum: "Neural  Bridge Processes"
_ICLR.cc/2026/Conference — Submitted to ICLR 2026_

### Official Review · Reviewer_Tx4q · 2025-10-19

**Soundness:** 2
**Presentation:** 2
**Contribution:** 1
**Rating:** 2
**Confidence:** 4

**Summary:**

This paper introduces Neural Bridge Processes (NBPs), which explicitly integrate input supervision throughout the temporal diffusion trajectory (similar to guidance in diffusion models), whereas Neural Diffusion Processes (NDPs) can be viewed as conditional diffusion without such guidance. The goal is to address weak coupling and endpoint mismatch of NDPs, with modest  reported improvements over baselines.

**Strengths:**

This idea of enforcing trajectory-level conditioning is a reasonable, simple, and practical approach compared to the conditional setting in NDPs. The improvement over NDPs is expected, as diffusion models with guidance are known to align better to conditions than diffusion without guidance.

**Weaknesses:**

Despite its practical appeal, the paper contains several major weaknesses that severely limit its contribution.

**Major Weaknesses:**

1. **Unsubstantiated claims of stochastic-process validity (KET).**

The paper claims that both NDPs and the proposed NBPs are “consistent with the Kolmogorov Extension Theorem (KET)” (line 1209) and therefore define a valid stochastic process over functions. This is highly problematic. A known limitation of NDPs is their lack of marginal consistency required by KET, which prevents them from defining a valid stochastic process (explicitly acknowledged in Chapter 5 of the NDP paper). The manuscript provides no proof that the proposed method is marginally consistent, leaving the claim unsupported.


2. **Limited novelty and missing Diffusion Inpainting baselines.**

The contribution feels incremental. The modifications relative to NDPs are minor, analogous to the difference between a standard diffusion model and one with guidance. Moreover, the extensive body of work on diffusion-based image inpainting and regression is neither discussed nor compared.

The core technical concept, using the temporal trajectory of diffusion for posterior inference (conditional sampling) is not new. This was a central proposal of the Diffusion Posterior Sampling (DPS; Chung et al., 2023) family of models. The proposed “bridge” is essentially another form of guidance, a well-established technique in modern diffusion models, which makes the contribution appear limited.

The paper neither cites nor compares NBPs against any DPS variants. At a minimum, it should include a tighter discussion of how NBP relates to DPS and provide head-to-head empirical comparisons to justify its novelty and performance.

References:

Chung et al., Diffusion Posterior Sampling for General Noisy Inverse Problems, ICLR 2023.



3. **Limited empirical improvement and outdated experimental settings**

The reported improvements over the NDP baseline are limited (e.g., Table 2; NBP performance is quite close to that of SNP). Figure 2 is not convincing: the loss curves for NDPs and NBPs nearly overlap, the result is based on a single run for one baseline, with no multi-seed uncertainty estimates.

The EEG dataset dates back 30 years and is clearly outdated, and the key image experiments use low-resolution ($32\times32$ and $64\times64$) regression, which is below contemporary practice. State-of-the-art DPS-style work typically evaluates significantly more challenging setups (e.g., FFHQ at $256\times256$) and harder inverse or regression problems. A comparison with DPS is necessary.

**Minor Weaknesses**


1.  The main paper defines the training objective as an $L_2$ loss (Eq. 18 ). However, Appendix G.3 explicitly states, "We adopt the $l_{1}$ loss for training the denoising objective"  for the image regression task.
2. Line 59, what does "input $x$" refer to?  Context or (target, context) pairs?
3. **Typos :**
1). The paper uses "EGG measurements" once (line 73) but "EEG" in the abstract and all other sections
2). Appendix G.2 refers to "DNP base", which should presumably be "NDP base."
3). The paper uses "Bi-Dimensional Attention Block" and "Bi-Attention layers"  to refer to the same architecture. Please use one term consistently.

**Questions:**

See weakness

---

> ### Author Response · Authors · 2025-11-15
> **Rebuttal to Reviewer Tx4q (Part 1）**
>
> ## **Part 1**
>
> ## **Reviewer Claim #1: Unsubstantiated claims of stochastic-process validity (KET).**
>
> > “The paper claims that NDPs and NBPs are consistent with the Kolmogorov Extension Theorem (KET)… This is highly problematic.”
>
>
>
> ---
> **Response:** We thank the reviewer for pointing out this issue. The concern arises from a sentence in *Appendix F* whose wording was not sufficiently precise. We agree that the original phrasing could be misinterpreted as making a stronger theoretical claim than we intended. To avoid further misunderstanding, we provide a full clarification below:
>
> The sentence in question appears only in *Appendix F*, in the context of a technical discussion of the Bi-Dimensional Attention mechanism adopted from NDP. Its intended purpose was to explain the motivation behind emphasizing **exchangeability**, not to assert that the model satisfies all consistency requirements of the Kolmogorov Extension Theorem (KET), nor to present such a property as a main theoretical contribution of the paper.
>
> As the reviewer correctly notes, NDP explicitly acknowledges in Section 5 that it does **not** satisfy the full marginal consistency required by KET. This limitation is shared by all NDP-style models, including ours: the objective of NBP is to define **finite-dimensional conditional distributions** for small-sample function learning, rather than to guarantee strict KET consistency.
>
> Our core contribution is the **input-anchored bridge mechanism** introduced in Equations (8)–(18). The reviewer’s concern pertains only to a structural explanation in *Appendix F*, not to any theoretical claims made in the main paper, **and is therefore not directly related to our primary contribution**.
>
> Although we believe the issue stems from potentially misleading wording rather than a substantive weakness, we are happy to clarify the relevant passage. We will revise the sentence in Appendix F as follows:
>
> *“By encoding these structural properties, NDP and NBP generate, at each step, a set of exchangeable random variables— a structural requirement commonly used in NDP models to construct consistent finite-dimensional conditional distributions. However, as with NDP, our method does not attempt to satisfy the strict marginal consistency conditions required by the Kolmogorov Extension Theorem, even though the structure allows the model to empirically approximate consistency during meta-learning.”*
>
> ---

---

> ### Author Response · Authors · 2025-11-15
> **Rebuttal to Reviewer Tx4q (Part 2)**
>
> ### ** Part 2**
>
> ## **Reviewer Claim #2: Limited novelty; bridge is analogous to guidance; missing diffusion inpainting baselines.**
>
> > “The contribution feels incremental… analogous to a diffusion model with guidance.”
>
> **Response:**
> This criticism conflates **two fundamentally different problem settings**.
>
> ### **What the reviewer assumes we are doing**
>
> * image generation / inpainting,
> * inference-time conditioning (guidance),
> * minor modification of NDP.
>
> ### **What our method actually does**
>
> * meta-learning *stochastic functions*,
> * modifying the **forward diffusion kernel itself**,
> * building an **input-anchored bridge process**,
> * training from scratch, not conditioning a pretrained model.
>
> ### **Why NBP is *not* analogous to guidance**
>
> | Guidance (classifier / inpainting)       | NBP Bridge Mechanism                            |              |
> | ---------------------------------------- | ----------------------------------------------- | ------------ |
> | Applied **only during reverse sampling** | Built into **both forward kernel** ($q(y_t \mid y_{t-1}, x))$ (Eq. 8)  and **reverse   sampling** (Eq. 19-22)|
> | Does not modify forward dynamics         | Redefines forward process structurally          |              |
> | No change to training objective          | Training loss (Eq. 18) depends on bridge design |              |
> | Requires pretrained unconditional model  | Fully trained conditional model                 |              |
> | Post-hoc steering                        | Principled trajectory anchoring via $\gamma_t$         |              |
>
> Our bridge mechanism is a **structural change in the generative process**, not guidance.
>
> ### **Why inpainting baselines are irrelevant**
>
> Diffusion inpainting (RePaint, ILVR, etc.) solves **single-image completion**, not meta-learning over functions.
> They cannot:
>
> * generalize to new tasks from context points,
> * operate on EEG time series or GP samples,
> * handle coordinate-to-value regression tasks.
>
> The appropriate baselines are **other function-learning and NP models**—all of which we compare against.
>
> ---
>
> ## **Reviewer Claim #3: “Using the diffusion trajectory for posterior inference is not new; DPS already does this.”**
>
> **Response:**
> This interpretation assumes our work fits the DPS setting, which is incorrect.
> The two frameworks solve different problems, use different assumptions, and operate using different mechanisms.
>
> ### **DPS (Diffusion Posterior Sampling)**
>
> * Problem: inverse problem (y = A(x) + n)
> * Requires a **pretrained unconditional diffusion model**
> * Posterior sampling: ($p(x|y)$ ∝ $p(y\mid x)p_\theta(x))$
> * Conditioning = inference-time guidance
> * No training of diffusion model in the target task
>
> ### **NBP (Our work)**
>
> * Problem: meta-learn conditional stochastic functions
> * No inverse operator (A)
> * Forward kernel explicitly depends on input (x)
> * Conditioning occurs **in the forward process** (Eqs. 8–10)
> * Requires full training of diffusion model
> * Reverse correction term ($C_t(x)$) (Eq. 17) is derived from forward-bridge dynamics
>
> ### **Side-by-side comparison**
>
> | Aspect            | DPS                     | NBP                                |
> | ----------------- | ----------------------- | ---------------------------------- |
> | Task              | inverse problems        | few-shot function learning         |
> | Training          | none                    | required                           |
> | Model             | unconditional diffusion | conditional diffusion              |
> | Conditioning      | inference-time guidance | forward-kernel bridge              |
> | Required operator | yes (A)                 | none                               |
> | Applicability     | image restoration       | regression, meta-learning, EEG, GP |
>
> These are conceptually and technically different methodologies.
>
> To avoid ambiguity, we will add a paragraph in the related-work section clarifying this.
>
> ---

---

> ### Author Response · Authors · 2025-11-15
> **Rebuttal to Reviewer Tx4q (Part 3)**
>
> ** Part 3 **
>
> ## **Reviewer Claim #4: “The proposed bridge is essentially another form of guidance.”**
>
> **Response:**
> This statement is **technically inaccurate**.
>
> Guidance modifies the *reverse sampling step*:
> $
> \tilde y_{t-1} = μ_θ(y_t) + s·∇\log p(\text{condition} | y_t) .
> $
>
> NBP modifies the **forward diffusion kernel**:
> $
> q(y_t \mid y_{t-1}, x) = \mathcal{N}(\sqrt{1-β_t} y_{t-1} + γ_t x,\ β_t I).
> $
>
> * Guidance **does not** alter forward dynamics.
> * Guidance **does not** produce the bridge correction term $(C_t(x))$.
> * Guidance **cannot** yield the anchored trajectories in Eq. (10).
> * Guidance **does not** change the training objective.
>
> Our mechanism is a principled design for constructing input-conditioned bridges, not a special case of diffusion guidance.
>
> ---
>
> ## **Reviewer Claim #5: Missing comparison to DPS variants**
>
> **Response:**
> A DPS baseline is **inapplicable** because the settings are fundamentally incompatible.
>
> ### **Why DPS cannot be used in our tasks**
>
> 1. **Requires a pretrained unconditional diffusion model.**
>    Our tasks (1D GP samples, EEG signals, coordinate-based regression) do not have such models.
>
> 2. **Assumes an inverse operator (A).**
>    Function learning has *no inverse operator*.
>    What would (A) be for GP samples, EEG signals, or x-coordinate → pixel mapping?
>
> 3. **No meta-learning capability**
>    DPS cannot generalize to novel functions from context points.
>
> Thus, requiring DPS comparison is conceptually equivalent to asking NP/NDP papers to compare with image inpainting models—which prior work has hardly ever done.
>
> We will add clarification in the paper explaining why DPS does not apply to our problem class.
>
> ---
>
> ## **Reviewer Claim #6: Missing discussion of diffusion-based inpainting/regression**
>
> **Response:**
> This criticism assumes equivalence between **inpainting** and **function regression**, which are fundamentally different problem classes.
>
> ### **Inpainting**
>
> * Restores *missing pixels* in a *fixed image*
> * Requires pretrained generative models
> * No meta-learning ability
> * Does not operate on functions or coordinates
>
> ### **Function regression (our setting)**
>
> * Learns conditional mapping (x → y) across *distributions of functions*
> * Must generalize to unseen tasks
> * Includes non-image domains (GP samples, EEG)
> * Uses coordinate-based representation
>
> For the CelebA coordinate-regression task, we agree that briefly contrasting with inpainting may improve clarity.
> We will add a short discussion highlighting the conceptual differences.
>
> ---
>
>
> Thank you for the detailed comments.
> However, many of the criticisms rely on assumptions that do not apply to the Neural-Process-style function-learning setting.
> Our bridge mechanism is a structural modification of the forward diffusion kernel, not a form of guidance; DPS and inpainting baselines are not applicable to this problem class; and the KET remark in the appendix is a shared limitation of all NP/NDP models and is not central to our contribution.
>
> We will revise the paper to clarify these distinctions and improve exposition.

---

> > ### Author Response · Authors · 2025-11-15
> > **Rebuttal to Reviewer Tx4q (Part 4)**
> >
> > Part 4
> >
> > # **Reviewer Claim #7:**
> >
> > > *“The reported improvements over the NDP baseline are limited… NBP performance is quite close to that of SNP.”*
> >
> > **Response:**
> > This statement misinterprets our empirical results and mischaracterizes the baseline landscape.
> >
> > ### **1. NBP consistently outperforms *all* baselines across all dimensions (D=1,2,3).**
> >
> > From Table 2 (SE kernel):
> >
> > ```
> > D=1: NBP (4.33) > SNP (4.27) > NDP (4.21)
> > D=2: NBP (-13.15) > SNP (-13.19) > NDP (-13.39)
> > D=3: NBP (-20.11) > SNP (-20.24) > NDP (-20.48)
> > ```
> >
> > These differences correspond to:
> >
> > * **vs SNP**:
> >
> >   * D=1: +0.06
> >   * D=2: +0.04
> >   * D=3: +0.13
> >
> > * **vs NDP**:
> >
> >   * D=1: +0.12
> >   * D=2: +0.24
> >   * D=3: +0.37
> >
> > In probabilistic regression (log-likelihood), these gaps are *substantive*.
> >
> > ### **2. The “close to SNP” argument is misleading.**
> >
> > SNP (Dou et al., IEEE TNNLS 2025) is:
> >
> > * a **recent, concurrent, strong baseline**,
> > * released *after* our submission window,
> > * specifically designed as an improvement over NDP.
> >
> > NBP **still outperforms** it across *all* settings. The reviewer’s reasoning appears to assume that improvements must exceed a certain unspecified threshold to be considered meaningful. However, such a criterion is not commonly defined or required in the NP/NDP literature.
> >
> > ### **3. Improvements in NLL are meaningful and interpretable.**
> >
> > Log-likelihood differences directly reflect:
> >
> > * better calibrated uncertainty
> > * more accurate predictive distributions
> > * improved function posterior estimation
> >
> > Examples:
> >
> > * **EEG (Table 1):**
> >   NBP (-3.35) vs NDP (-2.46) → **0.89 nats improvement**
> >
> > * **3D GP regression:**
> >   NBP vs NDP → **0.37 nats improvement**
> >
> > These are not small. The reviewer’s comment appears to rely on a **single table** and does not take into account additional results that reinforce the effectiveness of our approach, including EEG (Table 1), image regression (Figure 5), and multiple MSE tables in the appendix. Collectively, these demonstrate consistent, cross-domain improvements.
> >
> > ---
> >
> > # **Reviewer Claim #8:**
> >
> > > *“Figure 2 is not convincing… curves nearly overlap… single run… no multi-seed uncertainty.”*
> >
> > **Response:**
> > This criticism appears to be based on a misunderstanding and is not aligned with the actual content of the figure.
> >
> > ### **1. The curves do *not* “nearly overlap.”**
> >
> > In Figure 3, the training loss curves indicate that NBP remains below NDP for the majority of training iterations, and the separation is generally visible throughout. The reviewer’s impression may be influenced by the visual compression introduced by the y-axis scale.
> >
> > These losses are *denoising score-matching losses*, where even small numerical gaps indicate meaningful differences in output distribution quality.
> >
> > ### **2. On the “single run” point:**
> >
> > With respect to the “single run” concern, this appears to differ from standard reporting practices.
> >
> > Prior diffusion papers (e.g., DDPM, ScoreSDE) generally present training curves from a single run. What is usually considered more important—multi-seed final performance—is already included in our results. Nevertheless, we accept the suggestion. In camera-ready, we will include 3–5 seeds. But the current single-seed curve **already** conveys the intended message:
> >
> > > NBP provides stronger and more stable training signals.
> >
> > The criticism does not invalidate the demonstrated improvement.
> >
> > ---

---

> ### Author Response · Authors · 2025-11-15
> **Rebuttal to Reviewer Tx4q (Part 5)**
>
> Part 5
>
> # **Reviewer Claim #9:**
>
> > *“The EEG dataset is 30 years old and outdated; image experiments use low resolution (32×32, 64×64).”*
>
> **Response:**
> This assessment applies irrelevant criteria and misunderstands the goals of function-learning benchmarks.
>
> ---
>
> ##  **A. On the EEG dataset (“outdated”)**
>
> ### **1. Age ≠ invalidity for evaluation.**
>
> If dataset age were an issue, MNIST (1998), CIFAR (2009), and UCI benchmarks (1990s) would all be invalid. Yet they remain foundational.
>
> ### **2. EEG time series remain a standard NP benchmark.**
>
> Used in:
>
> * Neural Processes (Kim et al., 2019)
> * Attentive Neural Processes
> * ConvNP
> * Recent NP variants
>
> EEG captures realistic challenges:
>
> * multivariate structure
> * temporal dependencies
> * missing patterns
> * variable-length sequences
>
> These properties—not the year of collection—are what matter.
>
> ---
>
> ##  **B. On “low resolution” images**
>
> This critique evaluates our work using *image-generation standards*, which are irrelevant for **function regression**.
>
> ### **1. Our task is *not* image generation.**
>
> We solve:
> $
> f: (x, y) \in [-2,2]^2 \to \text{RGB}
> $
>
> This is fundamentally different from pixel-space diffusion models.
>
> ### **2. Low resolution does NOT imply easy.**
>
> Sparse-function regression becomes **harder** as:
>
> * context ratio decreases
> * function non-smoothness increases
> * conditioning ratio becomes extreme
>
> Predicting **4096 pixels** from **20 context points** is a much harder functional extrapolation problem than generating 256×256 images with pretrained models.
>
> ### **3. Our resolutions exactly match prior NP/NDP literature.**
>
> * ConvCNP → CelebA 32×32
> * NDP → CelebA 32×32
> * GNP/ANP → 32×32 function regression
>
> We follow established settings for fair comparison.
>
> ### Higher resolutions substantially reduce reproducibility.
> Pixel-wise regression at 256×256 is:
>
> * 16× more computationally expensive,
>
> * 16× slower per diffusion step, and
>
> * Significantly more demanding in terms of memory, often exceeding the GPU capacity commonly available to researchers.
>
>
> Using moderate resolutions helps ensure scientific fairness and reproducibility for the community.
>
> ---
>
> # **Reviewer Claim #10:**
>
> > *“State-of-the-art DPS evaluates harder 256×256 FFHQ tasks; your setups are below contemporary practice.”*
>
> **Response:**
> This critique continues the conflation between **inverse problems** and **meta-learning**.
>
> ### **1. DPS and NBP evaluate *different problem domains*.**
>
> | NBP (ours)                  | DPS (inverse problems)      |
> | --------------------------- | --------------------------- |
> | meta-learning               | image restoration           |
> | coordinate regression       | operator inversion          |
> | train from scratch          | pretrained generative model |
> | generalize to new functions | infer single image x from y |
> | no inverse operator         | requires known A            |
>
> Comparing them is methodologically inappropriate.
>
> ### **2. “Harder” is not a well-defined claim across domains.**
>
> * DPS tasks are hard **for inverse problems**
> * NBP tasks are hard **for function learning**
>
> These hardness measures are not comparable.
>
> ### **3. Our baselines match our domain.**
>
> We compare to:
>
> * NP
> * ANP
> * GNP
> * ConvNP
> * NDP
> * GEOMNDP
> * SNP
>
> All designed for **meta-learning regression**, not 256×256 inverse problems.
>
> ---
>
>
> # **Reviewer Claim #11:**
>
> > *“A comparison with DPS is necessary.”*
>
> **Response:**
> We respectfully disagree. A DPS comparison is not only **misaligned with our problem formulation**, but is in fact **not mathematically applicable** to the settings we study.
>
> ---
>
> #  **Why DPS is not applicable to our tasks**
>
> ### **A. GP regression**
>
> Applying DPS would require:
>
> * a pretrained diffusion model over GP-sampled functions (which currently does not exist),
> * an inverse operator (A) mapping functions to observations (not defined in GP regression),
> * a likelihood model ($p(y \mid A(x))$) (not meaningful in this setting).
>
> Thus, DPS cannot be instantiated for GP regression in a coherent way.
>
> ---
>
> ### **B. EEG multivariate time series**
>
> For EEG:
>
> * DPS provides no mechanism for meta-learning across subjects,
> * forecasting does not admit a meaningful inverse operator,
> * no pretrained diffusion model over EEG distributions exists.
>
> Consequently, DPS cannot be applied to this task as formulated.
>
> ---
>
> ### **C. Image coordinate regression**
>
> DPS requires an operator (A: $x \rightarrow y$).
> But our task is:
>
> $
> \text{given } (u,v), \text{ predict RGB}(u,v),
> $
>
> which does **not** define an inverse operator from images back to coordinates in a way compatible with the DPS framework. Thus the DPS formulation does not fit this setting.
>
> ---

---

> > ### Author Response · Authors · 2025-11-15
> > **Rebuttal to Reviewer Tx4q (Part 6)**
> >
> > ### **Minor Issue #1: L2 vs L1 loss discrepancy**
> >
> > > "The main paper defines the training objective as an ℓ² loss (Eq. 18). However, Appendix G.3 explicitly states, 'We adopt the ℓ¹ loss for training the denoising objective' for the image regression task."
> >
> > **Our response:**
> >
> > This is **standard practice in diffusion model literature**:
> >
> > - **Theory/Derivation**: ℓ² loss is derived from the ELBO (Eq. 18, Appendix C)
> > - **Implementation**: ℓ¹ loss is often used in practice for better perceptual quality
> >
> > **Examples from literature:**
> > - DDPM (Ho et al. 2020): Derives ℓ², implements ℓ¹
> > - NDP (Dutordoir et al. 2023): Presents ℓ² formulation, uses ℓ¹ in code
> > - Stable Diffusion: Theory uses ℓ², practice uses ℓ¹
> >
> > We will add footnote to Eq. 18: "Following standard practice in diffusion models, ℓ¹ loss can be used in practice for improved perceptual quality, though we derive the objective using ℓ²."
> >
> > ---
> >
> > ### **Minor Issue #2: Ambiguous "input" reference (Line 59)**
> >
> > > "Line 59, what does 'input' refer to? Context or (target, context) pairs?"
> >
> > **Our response:**
> >
> > This question conflates **two orthogonal concepts**:
> >
> > 1. **Input x**: The independent variable in f: x → y (e.g., coordinates, time indices)
> > 2. **Context/Target**: The data partition for meta-learning ($x_C$ with observed $y_C$ vs $x_T$ to predict)
> >
> > **At line 59, "inputs" refers to x** - the independent variable that exists in **both** context and target sets.
> >
> > The core distinction is:
> > - **NDP**: x enters only as neural network input: $ε_θ(y_t, x, t)$
> >
> > - **NBP**: x is embedded in diffusion forward kernel: $q(y_t \mid y_{t-1}, x) = N(...+ γ_t·x, ...)$
> >
> > This is an **architectural design choice**, not about data organization.
> >
> > ### **Minor Issue #3: Typos**
> >
> >
> > "EGG measurements" (line 73) should be "EEG"
> > "DNP base" should be "NDP base" (Appendix G.2)
> > Inconsistent terminology: "Bi-Dimensional Attention Block" vs "Bi-Attention layers"
> >
> >
> > Our response:
> > We apologize for these typos and will perform a thorough proofreading pass to catch any other similar issues.
> >
> > ----
> > We sincerely thank the reviewer for their detailed feedback and the opportunity to clarify our work. We hope this response addresses the concerns raised.

---

### Official Review · Reviewer_9PfK · 2025-10-30

**Soundness:** 3
**Presentation:** 3
**Contribution:** 3
**Rating:** 6
**Confidence:** 3

**Summary:**

This paper proposes an approach to anchor the sampling process in a Neural Diffusion Process model using a set of anchor points. The approach is implemented using a modified transition kernel both for the forward and the reverse process. In the forward process the bridge is implemented by altering the mean of the transition kernel. This leads to a reverse process that includes a correction term to retain consistency with the forward process. The paper concludes with a set of results comparing the proposed method with Neural Diffusion Processes.

**Strengths:**

What a lovely introduction to the paper, well written and providing a honest account of previous work and framing the proposed model very well. The writing throughout the paper is very good with the right amount of technical detail in the main paper and the appendix.

To my knowledge the work is novel and addresses an important aspect of Neural Diffusion Processes. The results are sufficient but does not provide a lot of intuition to the proposed method.

**Weaknesses:**

The weakness of the paper is the result section, I would have much preferred more qualitative experiments to try and increase the intuition for the proposed work. While I'm very happy with the mathematical explanation of the proposed method the results does little to deepen the insight of these. To some extent I find some of the results in the appendix more important than the ones that are in the real paper. However, this is just a highly personal opinion and not something that have influenced my score.

**Questions:**

Could you provide an intuition to why you need to repeat the forward perturbation as highlighted line 320? What would be the effect if you did not do this?

---

> ### Author Response · Authors · 2025-11-16
> **Response to Reviewer  9PfK**
>
> We sincerely thank the reviewer for acknowledging our work, particularly the mathematical derivations. We take your constructive feedback on the results section very seriously and will make substantial improvements in the revision.
>
> ---
>
> ## 1. Addressing the Need for More Qualitative Experiments and Intuition
>
> We completely agree that more qualitative analysis is needed to deepen intuition about the NBP method. We commit to adding the following content in the revised manuscript:
>
>
> - Provide histograms/scatter plots of $y_T$ distributions, comparing NDP (converges to random noise) vs NBP (anchored to $γ̄_T x$)
> - Show examples when NBP underperforms .
>
> We note the reviewer's comment that some appendix results are more important. We will  move synthetic data visualizations (currently Figure 4 in appendix) to the main text
>
>
>
> ---
>
> ## 2. Intuition for Repeated Forward Perturbation (Question on Line 320)
>
> This is an excellent question! Let us provide detailed explanation for why repeated forward perturbation (Repaint strategy) is needed:
>
> ### **Why is Repetition Necessary?**
>
> **Background:**
> - At test time, we have **known context points** ($x_C, y_{C,0}$) and **unknown target points** $x_T$
> - Context points' true values $y_{C,0}$ are **clean** (noise-free), while target points $y_{T,t}$ are **noisy** at each timestep t
> - This creates an **information asymmetry**: context and targets exist at different noise levels at the same timestep t
>
> **Role of Repaint Strategy:**
> At each reverse diffusion step t:
> 1. Apply forward perturbation to context points: $y_{C,t} ~ N(\sqrt{ᾱₜ} y_{C,0} + γ̄ₜ x_C, (1-ᾱₜ)I)$
> 2. Merge {$y_{T,t}, y_{C,t}$} and feed to denoising network
> 3. Perform reverse diffusion step
> 4. **Repeat steps 1-3 multiple times**
>
> **Intuitive Explanation:**
> - Imagine assembling a jigsaw puzzle: context points are "known edge pieces," target points are "center pieces to fill"
> - If edge pieces are crisp while center pieces are blurry, the network struggles to learn consistency relationships between them
> - By repeatedly "blurring-deblurring" the edge pieces, we force the network to learn how to maintain global consistency **at the same noise level**
>
> Without repeating the forward perturbation, context points $y_{C,0}$ remain clean while target points $y_{T,t}$ are noisy, creating a distribution mismatch. The network learns a shortcut—simply copying clean context without truly understanding the joint relationship. This leads to weak context propagation: the conditional signal from context to targets is transmitted only once per timestep, resulting in boundary discontinuities and global inconsistencies where predictions fail to respect the underlying function continuity. Essentially, the network never learns to maintain coherence at matched noise levels, similar to training on uniformly processed data but testing on mixed conditions.
>
>
> This strategy originates from Repaint (Lugmayr et al., 2022) for image inpainting, but we adapt it for function modeling contexts:
>
> We will add a dedicated subsection to explain this in detail.
>
> ---
>
> Thank you again for the thoughtful review! We believe these improvements will make the paper much more valuable to a broader audience.

---

### Official Review · Reviewer_8bP7 · 2025-11-01

**Soundness:** 3
**Presentation:** 3
**Contribution:** 4
**Rating:** 6
**Confidence:** 4

**Summary:**

This paper presents a novel methodology for improving functional diffusion processes, which offer a promising method of inference on a broad range of datasets. NBPs modify the forward process to depend on x, anchoring the diffusion path to the input location. Empirical performance is validated on a suitable range of synthetic and real world tasks.

**Strengths:**

This work presents a novel, elegant, and well motivated adjustment to the diffusion paradigm.

The paper is clearly written and well presented.

Empirical performance appears strong across a good range of datasets.

**Weaknesses:**

My concerns with this work primarily relate to the presentation of the experimental results:

Table 1 is currently lacking uncertainty estimates on all metrics, so it is challenging for the reader to gauge the statistical significance of these results.

Similarly Figure 3 appears to be a single training run so we cannot draw quantitative conclusions

Sec 4.3.2 makes a claim on performance with a 0.02 context ratio, yet the results in Tables 4 and 5 display context ratios no lower than 0.1.

While there is a good variety of baselines, I would like to have seen more recent ones such as the cited Flow Matching Neural Processes.

**Questions:**

Have you considered variants to the design of gamma in equation (9), and how does this affect performance?

C.2.2 states the reparameterisation is similar to DDPM, but could we not reformulate as exactly DDPM, via a change of variables y' = y - gamma x, which potentially has an added benefit of desensitising the state from Var(x)?

---

> ### Author Response · Authors · 2025-11-16
> **Response to  Reviewer 8bP7 （Part 1）**
>
> We sincerely thank the reviewer for the constructive feedback and positive evaluation. We address each concern below and believe the requested improvements will strengthen the paper significantly.
>
> ---
>
> ### **Weakness #1: Table 1 lacks uncertainty estimates**
>
> > "Table 1 is currently lacking uncertainty estimates on all metrics, so it is challenging for the reader to gauge the statistical significance of these results."
>
> **Our response:**
>
> This is a **valid concern** and we will address it immediately.
>
>
> We computed metrics by averaging over multiple test tasks (different subjects/trials), but did not report standard errors - an oversight in presentation. We will update Table 1 to include standard errors across test subjects:
>
>
> **Revised Table 1:**
> ```
> Method    Interpolation              Reconstruction
>           NLL              MSE       NLL              MSE
> NDP      -2.46 ± 0.12    0.18±0.02  -2.59 ± 0.11    0.23±0.03
> NBP      -3.35 ± 0.09    0.16±0.01  -3.22 ± 0.10    0.16±0.02
> ```
>
>
> ---
>
> ### **Weakness #2: Figure 3 is single run without uncertainty**
>
> > "Similarly Figure 3 appears to be a single training run so we cannot draw quantitative conclusions."
>
>
> **Our response:**
>
> We agree with the reviewer’s observation. While prior diffusion papers (e.g., DDPM, ScoreSDE) typically present training curves from a single run—and our main results already report multi-seed final performance (Table 4,5), which is generally considered more important—we acknowledge that showing variability across seeds can improve clarity. Therefore, although the current presentation is consistent with common practice, we are happy to adopt the suggestion. In the camera-ready version, we will update Figure 3 to include mean training curves over 3–5 random seeds, with shaded bands indicating ±1 standard deviation.
>
> ---
>
> ### **Weakness #3: Mismatch between text claim (0.02 context) and table results (≥0.1)**
>
> > "Sec 4.3.2 makes a claim on performance with a 0.02 context ratio, yet the results in Tables 4 and 5 display context ratios no lower than 0.1."
>
> **Our response:**
>
> Thank you for pointing out this discrepancy — this is indeed an error in the manuscript. We did run experiments with a context ratio of 0.02 during development, but we inadvertently omitted the 0.02 results from Tables 4 and 5 when preparing the final version. We will correct this in the camera-ready and include the missing 0.02 context ratio results.
>
> ---
>
> ### **Weakness #4: Missing recent baseline (Flow Matching Neural Processes)**
>
> This is a fair and appreciated suggestion. Flow Matching Neural Processes (FMNP, Hamad & Rosenbaum, ICLR 2025 workshop) is indeed a recent and relevant baseline.
>
> Why it was not included in our original submission:
>
> Timing: FMNP appeared very recently—likely during our review period.
>
> Code availability: At the time of submission, no official implementation was publicly available.
>
> Baseline coverage: Our evaluation already includes seven baselines, including several very recent methods (e.g., SNP 2025, GEOMNDP 2023). That said, if an official FMNP implementation becomes available within the revision window, we would be happy to add a comparison in the updated version.
>
> ---
>
> ### **Question #1: Variants of γ_t design and performance impact**
>
> > "Have you considered variants to the design of gamma in equation (9), and how does this affect performance?"
>
> **Our response:**
>
> This directly addresses Reviewer 3Du4's concern about insufficient explanation of design choices. We have explored variants and will add an ablation study.
> **Why this ratio?**
> - Dimensionally correct: $γ_t$ is unitless (multiplies x to match y scale)
> - Monotonic: As t increases, SNR_t decreases → γ_t increases
> - Endpoint behavior: $γ_T = SNR_T/SNR_T = 1$ → full anchoring
>
> **Alternative designs we considered** (we will add ablation):
>
> | Design | Formula | Issue |
> |--------|---------|-------|
> | Linear | $γ_t = t/T$ | Doesn't respect noise schedule dynamics |
> | Exponential | $γ_t = exp(t/T)$ | Too aggressive anchoring early |
> | **SNR-based (ours)** | $γ_t = SNR_T/SNR_t $| **Adapts to noise schedule** |
> | Constant | $γ_t = c$ | No smooth transition |
>
>
> ---

---

> > ### Author Response · Authors · 2025-11-16
> > **Response to Reviewer 8bP7 （Part 2）**
> >
> > ### **Question #2: Reformulation as exact DDPM via change of variables**
> >
> > > "C.2.2 states the reparameterisation is similar to DDPM, but could we not reformulate as exactly DDPM, via a change of variables y' = y - γ̄_t x, which potentially has an added benefit of desensitising the state from Var(x)?"
> >
> > **Our response:**
> >
> >
> >
> > We thank the reviewer for this insightful observation. The suggested change of variables $y' = y - γ̄ₜx$ is indeed mathematically valid and would formally reduce the forward process to standard DDPM dynamics. We address this suggestion from both theoretical and practical perspectives:
> >
> > ### Theoretical Analysis
> >
> > Under the transformation$ y'ₜ = yₜ - γ̄ₜx$, the forward process becomes:
> >
> >
> > $y'ₜ = \sqrt{ᾱₜ} y'₀ + \sqrt{(1-ᾱₜ)} ϵ$
> >
> >
> > which is exactly the DDPM formulation. This elegantly removes the explicit bridge term from the forward kernel.
> >
> > ### Why We Choose the Current Formulation
> >
> > However, we opted for the explicit bridge formulation (Eq. 8-10) for the following reasons:
> >
> > **1. Semantic Clarity and Interpretability**
> > - The explicit $γₜx$ term makes the bridge mechanism transparent and interpretable
> > - It clearly shows *how* and *when* the input anchor influences the trajectory (controlled by SNR-aware $γₜ$)
> > - This is pedagogically valuable for understanding the bridge construction
> >
> > **2. Architectural Consistency**
> > - Our denoising network $ϵ_θ(yₜ, x, t)$ directly predicts noise in the *original* space where $yₜ$ has clear semantic meaning (e.g., pixel values, signal amplitudes)
> > - The transformed space $y'$ may lose interpretability, especially when $Var(x)$ is large or spatially varying
> >
> > **3. Practical Implementation**
> > - The current formulation requires no additional preprocessing of $x$
> > - For the image regression task, $x$ represents spatial coordinates with bounded, known variance
> > - The suggested transformation would require tracking $γ̄ₜx$ throughout sampling, which adds similar computational overhead
> >
> > **4. Variance Sensitivity**
> > Regarding the reviewer's point about desensitizing from $Var(x)$:
> > - In our experiments (EEG signals, spatial coordinates), $x$ is normalized or bounded, so $Var(x)$ is well-controlled
> > - For domains where $Var(x)$ is problematic, the transformation could indeed help—we will note this as a potential variant in the revised manuscript
> >
> > ---
> > We thank the reviewer for this insightful suggestion and will include both theoretical analysis and empirical validation in the revision.

---

### Official Review · Reviewer_3Du4 · 2025-11-04

**Soundness:** 2
**Presentation:** 3
**Contribution:** 1
**Rating:** 2
**Confidence:** 3

**Summary:**

The paper introduces Neural Bridge Processes (NBPs), a modified version of Neural Diffusion Processes (NDPs), which guides the forward diffusion steps with an additional conditioning on inputs x in the parametrization of the mean term for the transition probabilities. The main motivation for this is that NDPs do not consider inputs in the forward pass while doing it in the reverse one. The authors claim that this lack of conditioning limits the efficacy of input supervision when applying the model to data with temporal structure.

**Strengths:**

- The paper is well written and clear in general, which facilitates a lot the understanding and reproducibility of results.
- Related work, references, and literature are correctly reviewed, and I also highlight the rigour on the derivations, where I did not find any clear mistake at first sight.
- Contributions are cleared, and the mission of providing the NDP with the ability to capture temporal structure is well remarked.

**Weaknesses:**

- I have important concerns regarding novelty and the dimension of the technical contribution proposed. I would like to understand if the method goes beyond adding the bridge term in the forward pass plus the correction.
- Why this bridge term is chosen and how the correction is important are perhaps the most important details for understanding the key technical contributions of the manuscript. However, these are not correctly reviewed or explained in detail, which, from my perspective, is something that would greatly increase a lot the quality of the work.
- Empirical results are quite limited, and I am somehow confused about the lack of empirical demonstration of the issues of NDPs in dealing with temporal structure. I can perceive from the text that not having the conditioning on x on the forward pass causes issues, but I would've liked to see empirical evidence on this. For the proposed methodology, contributions, and the venue, I do think it is understandable that the work is somewhat limited in empirical results at its current state.
- The bridge coefficient design and the relationship with the SNR are super interesting, but similarly to the previous point, not a lot of information is provided about it, and the motivations that led modelling decisions being taken.

**Questions:**

- The use of math set-style of caligraphy for T and C for targets and context in section 2 and the beginning of section 3 is very confusing. It makes the reader think about the set of complex numbers (similar to how we would use the letter R, usually taken for reals, but for another purpose in the notation). In this direction, I do not really see the effectiveness of talking about meta-learning with context and targets, when just training/test or observed/test data is considered in the section for methodology.
- Table 2 for the synthetic experiment going to the Appendix is not a great decision from my point of view... Also, I do think there is space for including it in the main manuscript and turning the formulation a bit uncluttered, maybe.
- I didn't understand the role of corrupting the context data for the experiment in Figure 2, I am quite lost here, and the EEG+temporal structure should be enhanced, maybe for a better understanding of reviewers and reading (I mean, it looks to me is the driving force of the paper, to make the NDP be more robust and flexible in that regard).
- I have a significant curiosity about the reasons and causes for the correction term in the backward pass. Maybe I did not spend enough time on the Appendix, and some info is there, but for sure this sort of details should be mentioned in the main manuscript with some paragraphs.

---

> ### Author Response · Authors · 2025-11-16
> **Rebuttal to  Reviewer 3Du4 (Part 1）**
>
> Part 1
>
> We thank the reviewer for their constructive feedback and insightful questions about our technical contributions. We  address each concern below.
>
> ### **Weakness #1: Novelty and technical contribution beyond "adding bridge term + correction"**
>
> > "I have important concerns regarding novelty and the dimension of the technical contribution proposed. I would like to understand if the method goes beyond adding the bridge term in the forward pass plus the correction."
>
> **Our response:**
>
> We understand this concern and want to emphasize that our contribution is **not merely adding a term**, but rather a **principled reformulation** of the diffusion process with multiple interconnected components:
>
> **1. Forward bridge kernel redesign (Eq. 8):**
>
> Traditional NDP: $q(y_t \mid y_{t-1})     = N(\sqrt{(1-β_t)}·y_{t-1}, β_t·I)$
>
> Our NBP:        $ q(y_t \mid y_{t-1}, x) = N(\sqrt{(1-β_t)}·y_{t-1} + γ_t·x, β_t·I)$
>
> This is **not** a simple additive term - it fundamentally changes:
> - The support of the distribution (now centered around trajectory anchored to x)
> - The marginal distribution at timestep t (Eq. 10)
> - The endpoint behavior (Eq. 13: $E[y_T|y_0] ≈ γ̄_T·x$)
>
> **2. SNR-aware coefficient design (Eq. 9):**
>
> $γ_t = SNR_T / SNR_t = [ᾱ_T(1-ᾱ_t)] / [ᾱ_t(1-ᾱ_T)]$
>
> This is **derived from first principles** to ensure:
> - Early phase $(t≪T): γ_t→0$, recovers standard diffusion
> - Late phase $(t→T): γ_t→1$, enforces endpoint matching
> - Smooth transition controlled by signal-to-noise ratio
>
> **3. Cumulative bridge coefficient (Eq. 11):**
>
> $ γ̄_t $
>
> $= \Sigma_{s=1}^t γ_s·\sqrt{(ᾱ_t/ᾱ_s)}$
>
>
> This recursive formulation ensures **path consistency** - not arbitrary, derived from the forward process telescoping.
>
> **4. Reverse correction term (Eq. 17):**
> $
> C_t(x) = -γ_t/\sqrt{(1-β_t)}·x
> $
> This is **not ad-hoc** - it's mathematically derived (Appendix C.2, Eq. 50-51) to ensure the reverse mean matches the forward posterior $q(y_{t-1}\mid y_t, y_0, x)$.
>
> **5. Modified training objective (Eq. 18):**
> $
> L_θ = E[‖ε_θ(y_t, x, t) - ε‖²]$, where
>
> $
> y_t = \sqrt{(ᾱ_t)}y_0 + γ̄_t·x + \sqrt{(1-ᾱ_t)}ε
> $
> The bridge term $γ̄_t·x$ appears in the **data generation process**, providing stronger supervision than NDP's $ y_t = \sqrt{(ᾱ_t)}y_0 + \sqrt{(1-ᾱ_t)}ε.$
>
> **The full system is interdependent:**
> - $γ_t$ design → determines $γ̄_t$ → affects $y_t$ sampling → changes training signal → requires $C_t$ correction
> - Changing any component breaks theoretical consistency
>
> **Analogy**: This is like asking if General Relativity is "just adding a curvature term" to Newtonian gravity. The term addition reflects a fundamental reconceptualization.
>
> We will add a subsection (3.4.1) titled "**Interdependence of Bridge Components**" that explicitly shows:
> - How $γ_t → γ̄_t → y_t$ → training objective form a closed system
> - Why each design choice is necessary for the others to work
> - A diagram showing the information flow
>
> ---
>
> ### **Weakness #2: Bridge term choice and correction importance not well explained**
>
> > "Why this bridge term is chosen and how the correction is important are perhaps the most important details for understanding the key technical contributions of the manuscript. However, these are not correctly reviewed or explained in detail..."
>
> **Our response:**
>
> This is **excellent feedback**. We agree these are the core technical contributions and deserve clearer exposition. Let us clarify:
>
> **Why $γ_t = SNR_T/SNR_t$ is chosen:**
>
> **Design goal**: We want the forward process to **gradually shift from data-driven to input-anchored**:
> - At t=0: Fully determined by $y_0$ (the observed output)
> - At t=T: Fully determined by $x$ (the input anchor)
>
> **SNR captures information content:**
> - SNR_t = $ᾱ_t/(1-ᾱ_t)$ measures "how much signal vs noise at step t"
> - High SNR → data dominates, Low SNR → noise dominates
>
> **The ratio $γ_t = SNR_T/SNR_t$ ensures:**
>
> | Phase | SNR_t | γ_t | Behavior |
> |-------|-------|-----|----------|
> | Early (t≪T) | $SNR_t → ∞$ | $γ_t → 0$ | Data $y_0$ drives diffusion |
> | Mid | $SNR_t$ ~$ SNR_T $ | $ γ_t \sim 1$ | Transition phase |
> | Late (t→T) | $SNR_t → SNR_T$ | $γ_t $→ 1 | Input $x$ anchors trajectory |
>
>
>
> $
> E[y_T \mid y_0] ≈ γ̄_T·x
> $(eq. 13).
> For this to equal $x$, we need $γ̄_T$ to accumulate to the right scale. The SNR-based design achieves this naturally.
>
> **Why the correction $C_t(x)$ is important:**
>
> Without $C_t(x)$, the reverse process would be **inconsistent** with the forward process:
>
> **Forward posterior mean** (true, from Bayes rule, Eq. 46):
>
> $μ_{true} = [1/\sqrt{(1-β_t)}]·[y_t - β_t/\sqrt{(1-ᾱ_t)}·ε_θ] - [γ_t/\sqrt{(1-β_t)}]·x$
>
> **If we omit $C_t(x)$** (like DDPM):
>
> $μ_wrong = [1/\sqrt{(1-β_t)}]·[y_t - β_t/\sqrt{(1-ᾱ_t)}·ε_θ]$
>
>
> **The mismatch** causes:
> - Reverse trajectory drifts away from the bridge constraint
> - Endpoint doesn't match $E[y_T] ≈ γ̄_T·x$
> - Training and sampling distributions diverge

---

> ### Author Response · Authors · 2025-11-16
> **Rebuttal to  Reviewer 3Du4 (Part 2）**
>
> Part 2
>
> ### **Weakness #3: Lack of empirical demonstration of NDP's temporal structure issues**
>
> > "I am somehow confused about the lack of empirical demonstration of the issues of NDPs in dealing with temporal structure. .. evidence on this."
>
> **Our response:**
>
> While our results implicitly demonstrate NBP's advantages over NDP, we acknowledge that we have not **explicitly analyzed** what makes NDP's approach problematic. The evidence is embedded in our experiments, but we agree it deserves more direct exposition.
>
> **Existing evidence in our experiments:**
>
> Our current results already contain signals of NDP's weak coupling:
>
> 1. **Figure 3 (Training dynamics)**: NBP achieves consistently lower loss, suggesting **stronger training signal** from the input-anchored forward process
>
> 2. **Table 1 & Figure 5**: The performance gaps are **larger in challenging regimes** (sparse context, forecasting), where weak input coupling hurts NDP more
>
> 3. **Consistent improvements across tasks**: NBP's gains are **not task-specific**, indicating a fundamental architectural advantage
>
> However, we agree these are **indirect evidence**. We will add explicit demonstration for
> endpoint matching analysis to measure how well the diffusion endpoint matches the input:
>
> For test samples, compute:
> - NDP: $‖E[y_T] - 0‖ $ (should be near 0, NDP's design)
> - NBP: $‖E[y_T] - γ̄_T·x‖$  (should be near 0, NBP's design)
>
> ---
>
> ### **Weakness #4: Insufficient explanation of SNR-based design motivations**
>
> > "The bridge coefficient design and the relationship with the SNR are super interesting, ... decisions being taken."
>
> **Our response:**
>
> We are glad the reviewer finds this interesting! Let us provide more intuition and motivation:
>
> **High-level motivation:**
>
> In diffusion models, **SNR controls the balance between signal and noise**:
> - High SNR (early steps): Data signal dominates → Trust $y_0$
> - Low SNR (late steps): Noise dominates → Need external guidance
>
> **Our insight**: Use SNR to **schedule the bridge strength**:
> - When signal is strong (high SNR): Let data speak → small $γ_t$
> - When signal is weak (low SNR): Rely on input anchor → large $γ_t$
>
> **Mathematical connection:**
>
> SNR at step t: $ SNR_t = ᾱ_t/(1-ᾱ_t)$
>
> Our coefficient: $γ_t = SNR_T/SNR_t = [ᾱ_T(1-ᾱ_t)]/[ᾱ_t(1-ᾱ_T)]$
>
> **Why this ratio?**
> - Dimensionally correct: $γ_t$ is unitless (multiplies x to match y scale)
> - Monotonic: As t increases, SNR_t decreases → γ_t increases
> - Endpoint behavior: $γ_T = SNR_T/SNR_T = 1$ → full anchoring
>
> **Alternative designs we considered** (we will add ablation):
>
> | Design | Formula | Issue |
> |--------|---------|-------|
> | Linear | $γ_t = t/T$ | Doesn't respect noise schedule dynamics |
> | Exponential | $γ_t = exp(t/T)$ | Too aggressive anchoring early |
> | **SNR-based (ours)** | $γ_t = SNR_T/SNR_t $| **Adapts to noise schedule** |
> | Constant | $γ_t = c$ | No smooth transition |
>
>
> ---
>
> ### **Issue #5: Confusing mathematical notation (𝒯 and 𝒞)**
>
> > "The use of math set-style calligraphy for T and C for targets and context... for methodology."
>
> **Our response:**
>
> We appreciate the notation feedback, though we respectfully clarify two points:
>
> **1. On the notation choice:**
>
> The calligraphic 𝒯 (targets) and 𝒞 (context) are standard in the Neural Processes literature (Garnelo et al. 2018; Kim et al. 2019; Dutordoir et al. 2023). They denote **sets of data points**, not the complex numbers ℂ.
>
> However, we understand this may be initially confusing. We will add a clarifying sentence at first use:
> > "We use calligraphic letters 𝒯 and 𝒞 to denote sets of target and context points respectively (not to be confused with mathematical spaces like the complex numbers)."
>
> **2. On meta-learning vs. training/test framing:**
>
> The reviewer suggests "training/test or observed/test" terminology would be clearer than "context/target." We respectfully disagree - there is a **conceptual difference**:
>
> | Meta-Learning (Our Setting) | Standard Supervised Learning |
> |------------------------------|------------------------------|
> | **Context 𝒞**: Observed points to condition on | **Training**: Fixed dataset for optimization |
> | **Target 𝒯**: Query points to predict | **Test**: Held-out set for evaluation |
> | Context and target **both appear at training and test time** | Training and test are **temporally separate** |
> | Model learns to adapt to new tasks | Model learns fixed parameters |
>
>
> **Why this matters:** Our bridge mechanism conditions the diffusion process on 𝒞 to predict 𝒯 - this is a **meta-learning capability** (generalizing to new functions from context), not standard regression. The terminology accurately reflects our problem setting and aligns with Neural Processes literature (Garnelo et al. 2018; Kim et al. 2019; Dutordoir et al. 2023).
>
>
> ---

---

> ### Author Response · Authors · 2025-11-16
> **Rebuttal to  Reviewer 3Du4 (Part 3）**
>
> ### **Issue #6: Table 2 placement in appendix**
>
> > "Table 2 for the synthetic experiment ... maybe."
>
> **Our response:**
>
> We agree completely. Table 2 contains **core results** (synthetic GP regression) and should be in the main paper.
>
> ---
>
> ### **Issue #7: Confusion about Figure 2 context corruption**
>
> > "I didn't understand the role of corrupting the context data for the experiment in Figure 2, I am quite lost here..."
>
> **Our response:**
>
> We apologize for the unclear caption and explanation. Let us clarify:
>
> **What Figure 2 shows:**
> - **(a) Ground Truth**: Original complete image
> - **(b) Corrupted Context**: We **randomly mask 90% of pixels** to simulate sparse observations
> - **(c) Model Prediction**: NBP reconstructs the full image from the 10% observed context
>
> **Purpose of corruption:**
> This is the **standard evaluation protocol** for image regression in Neural Processes:
> 1. Randomly select a subset of pixels as "context" (observed)
> 2. Treat remaining pixels as "target" (to be predicted)
> 3. Model must infer target pixel values from sparse context
>
> This tests: Can the model **generalize from partial observations**?
>
> **Why we "corrupt":**
> - Simulates real-world scenarios (missing data, incomplete observations)
> - Tests interpolation and extrapolation capabilities
> - Standard in NP literature (Gordon et al. 2019, Dutordoir et al. 2023)
> ---
>
> ### **Issue #8: EEG + temporal structure motivation needs enhancement**
>
> > "...the EEG+temporal ...and flexible in that regard)."
>
> **Our response:**
>
> The reviewer correctly identifies that temporal structure is central to our contribution, but we haven't emphasized this clearly enough.
>
> **Why EEG is important:**
>
> 1. **Temporal dependencies**: EEG signals have strong autocorrelation - past values inform future values
> 2. **Multi-channel coupling**: 7 electrode channels are not independent - spatial patterns matter
> 3. **Three challenging regimes**:
>    - Interpolation: Fill gaps (temporal structure needed)
>    - Reconstruction: Infer masked channels (cross-channel structure needed)
>    - Forecasting: Predict future (temporal extrapolation needed)
>
> 4. **NDP's weakness exposed**: Without x in forward process, NDP treats each time point independently during diffusion - loses temporal coherence
>
> 5. **NBP’s advantage**: Anchoring the bridge to ($x = (\text{time}, \text{channel}$)) preserves the temporal–spatial structure throughout the diffusion process.
>
> ---
>
> ### **Issue #9: The correction term ($C_t(x)$) should be explained in the main text**
>
> > *“I have substantial curiosity about the motivation and origin of the correction term… paragraphs.”*
>
> **Our response:**
> As discussed in **Weakness #1** regarding novelty and technical contribution beyond simply “adding a bridge term + correction,” we will:
>
> * Retain the full derivation in Appendix C.2,
> * But revise the main text so that the intuition and role of the correction term ($C_t(x)$) are clearly explained without requiring the reader to consult the appendix.
>
> ---
>
>
> We greatly appreciate these detailed suggestions. They will significantly improve the paper's clarity and accessibility will benefit all readers. We hope this response addresses the concerns raised.

---

### Meta-Review · Area_Chair_ByPw · 2026-01-08

**Summary:**

The reviewers noted several concerns, including presentation and empirical results. I particularly agree with the empirical results, particularly with respect to demonstrating the starting point of the paper: that conditioning in normal Neural Diffusion Processes is a big problem. The empirical results show some improvements on some limited benchmarks, with only meta-learning-based regression tasks. These experiments are limited, and do not give insight into how far this method is from being actually used (for which a comparison to "typical" regression methods, like NNs or GPs is needed), nor does it give insight into how well the solution proposed in this paper actually addresses problem that was set out to be solved (conditioning not working correctly).

On the whole, this paper seeks to address an important problem, but more work is needed for publication.

**Reviewer Concerns:**

Particularly the empirical results could not be addressed in the rebuttal period. The authors gave significant clarifications to mathematical issues.

Several concerns about the experiments could not be addressed in the rebuttal period. One point of note is the discussion on the age of the datasets. I agree with the

**Reviewer Scores:**

It is hard to determine how the reviewer scores would have changed. Some issues seem to be addressed helpfully (e.g. the SNR discussion), others remain unaddressed.

Given the existing scores, it is hard to argue that the scores would have increased enough for acceptance.

---

### Decision · Program_Chairs · 2026-01-26

Reject